# Shavenbaby and Yorkie mediate Hippo signaling to protect adult stem cells from apoptosis

Jérôme Bohère[1], Alexandra Mancheno-Ferris[1], Sandy Al Hayek[1,2,3], Jennifer Zanet[1], Philippe Valenti[1], Kohsuke Akino[4], Yuya Yamabe[4], Sachi Inagaki[5], Hélène Chanut-Delalande[1], Serge Plaza[1,6], Yuji Kageyama[4,5], Dani Osman[2,3], Cédric Polesello [1] & François Payre [1]

To compensate for accumulating damages and cell death, adult homeostasis (e.g., body fluids and secretion) requires organ regeneration, operated by long-lived stem cells. How stem cells can survive throughout the animal life remains poorly understood. Here we show that the transcription factor Shavenbaby (Svb, OvoL in vertebrates) is expressed in renal/nephric stem cells (RNSCs) of *Drosophila* and required for their maintenance during adulthood. As recently shown in embryos, Svb function in adult RNSCs further needs a post-translational processing mediated by the Polished rice (Pri) smORF peptides and impairing Svb function leads to RNSC apoptosis. We show that Svb interacts both genetically and physically with Yorkie (YAP/TAZ in vertebrates), a nuclear effector of the Hippo pathway, to activate the expression of the inhibitor of apoptosis *DIAP1*. These data therefore identify Svb as a nuclear effector in the Hippo pathway, critical for the survival of adult somatic stem cells.

[1] Centre de Biologie du Développement (CBD), Centre de Biologie Intégrative (CBI), Université de Toulouse, CNRS, Bat 4R3, 118 route de Narbonne, F-31062 Toulouse, France. [2] Faculty of Sciences III, Lebanese University, Tripoli 1300, Lebanon. [3] Azm Center for Research in Biotechnology and its Applications, LBA3B, EDST, Lebanese University, Tripoli 1300, Lebanon. [4] Department of Biology, Graduate School of Science, Kobe 657-8501, Japan. [5] Biosignal Research Center, Kobe University, 1-1 Rokko-dai, Nada, Kobe 657-8501, Japan. [6] Present address: Laboratoire de Recherche en Sciences Végétales (LSRV), CNRS, UPS, 24 chemin de Borde Rouge, Auzeville, 31326 Castanet-Tolosan, France. Correspondence and requests for materials should be addressed to C.P. (email: cedric.polesello@univ-tlse3.fr) or to F.P. (email: francois.payre@univ-tlse3.fr)

The family of OvoL/Ovo/Shavenbaby (Svb) transcription factors has been strongly conserved across evolution and is characteristic of animal species. Initially discovered in flies for a dual function in the development of the germline and of epidermal derivatives (Ovo/Svb)[1,2], mammalian orthologs (OvoL1-3) have soon been identified[3,4]. *OvoL/svb* genes produce several protein isoforms and the existence of three partially redundant paralogs in mammals complicates their genetic analysis. There is a single gene in *Drosophila*, which expresses germline- (*ovo*) and somatic-specific (*svb*) transcripts from different promoters. Previous work has well-established the role of Svb in the development of embryonic epidermal tissues[2], where it triggers a tridimensional cell shape remodeling for the formation of actin-rich apical extensions, called trichomes. *Svb* expression is driven by a large array of *cis*-regulatory regions, which have become a paradigm for elucidating the function[5,6] and evolution[7–9] of developmental enhancers. *Svb* enhancers directly integrate multiple inputs from upstream regulatory pathways[2,5,10] and often drive similar patterns[5,6,11], together conferring robustness to epidermal development in the face of varying environmental conditions and/or genetic backgrounds[5,6]. During embryogenesis, the Svb transcription factor directly activates a battery of >150 target genes[12–14] collectively responsible for actin and extra-cellular-matrix reorganization that underlies trichome formation. Recent studies have unraveled a tight control of the Svb protein activity in response to Polished rice peptides (Pri, also known as Tarsal-less), which belongs to a fast-growing family of peptides encoded from small open reading frames (smORF) hidden within apparently long noncoding RNAs[15,16]. Svb is first translated as a long-sized protein that acts as a repressor (Svb[REP])[17]. Pri smORF peptides then induce a proteolytic processing of Svb[REP] leading to the degradation of its N-terminal region and releasing a shorter activator form, Svb[ACT] [17,18]. Further work has demonstrated that *pri* expression is directly regulated by periodic pulses of steroid hormones[19], allowing a functional connection between hard-wired genetic regulatory networks (*svb* expression) and systemic hormonal control (*pri* expression) for a proper spatio-temporal control of epidermal cell morphogenesis[15].

Recent studies suggest that OvoL/Svb factors display broader functions throughout epithelial tissues in both normal and various pathological situations. Molecular profiling of human tumors has revealed that OvoL deregulation is a feature of many carcinomas, directly linked to the metastatic potential of morbid cancers[20–23], including kidney[24]. OvoL factors have been proposed[25,26] to counteract a conserved core of regulators composed of Snail/Slug and Zeb1-2 transcription factors, as well as the micro RNA *mir200*, well known to promote epithelial–mesenchymal transition (EMT)[27]. The activity of OvoL might help stabilizing a hybrid phenotype between epithelial and mesenchymal states[25], providing many advantages for both tumors and normal stem cells[28]. Indeed, recent data show that, like adult somatic stem cells, the most aggressive tumors often display a hybrid E/M phenotype[27], and the expression of specific OvoL isoforms can annihilate the metastatic potential of mammary tumors[20,29]. In addition, OvoL/Svb factors have been linked to the control of various progenitors/stem cells, from basal invertebrates[30] to humans[31–33]. Therefore, whereas a large body of evidence supports a key role for OvoL/Svb in the behavior of somatic stem cells, a functional investigation of their mode of action in vivo remains to be undertaken.

Here, we built on the knowledge and tools accumulated for the study of Svb function in flies to investigate its putative contribution to the behavior of somatic stem cells in the adult. We show that in Malpighian tubules, which ensure renal function in insects[34,35], *svb* is specifically expressed in the adult renal/nephric stem cells (RNSCs). We further find that the main function of Svb

in the kidney is to protect RNSCs from apoptosis by controlling the expression of the inhibitor of apoptosis, *DIAP1*, in interaction with Yorkie (Yki), a nuclear effector of the Hippo pathway.

## Results

**svb is expressed in RNSCs and controls their maintenance.** To assay whether *svb* might be expressed in the adult, we tested large genomic reporter constructs that cover each of the seven enhancers contributing to *svb* expression[7,8]. We found that one enhancer, *svb[E]* [8], drove specific expression in tiny cells of the Malpighian tubules (Supplementary Figure 1a, b).

Malpighian tubules are mainly composed of two types of differentiated cells[35]. The principal cells—characterized by the homeodomain Cut protein (Fig. 1a, b)—express the vacuolar-ATPase (V-ATPase) that establishes an $H^+$ electrochemical potential promoting *trans*-epithelial secretion of $Na^+$ and $K^{[+\ 34]}$. The second main population of Malpighian tubules are termed stellate cells, featured by the expression of the Teashirt (Tsh) transcription factor (Fig. 1a), and that regulate the transport of $Cl^-$ and water[34]. While both principal and stellate cells display large-sized polyploid nuclei, a third population of small diploid cells, originally referred to as tiny cells with putative myoendo-crine and/or neuroendocrine activity[36,37], are located in the lower tubules (Fig. 1a, b). Accumulated evidence now supports that these tiny cells ensure the renewal of at least some populations of kidney cells[38,39] and correspond to adult RNSCs (see also Supplementary Figure 4). It has been shown that RNSCs derive from a subpopulation of intestinal stem cell precursors that migrate in Malpighian tubules during post-embryonic development[40]. RNSCs are characterized by the expression of Escargot (Esg), a transcription factor of the Snail/SLUG family that is also expressed in intestinal stem cells[41] where it acts to prevent stem cell differentiation[42,43]. Co-localization with an *esg-LacZ* reporter confirmed that the *svb[E]* enhancer was active in RNSCs (Fig. 1c and Supplementary Figure 1). To define the minimal region of *svb* responsible for the expression in RNSCs, we assayed a collection of overlapping constructs[5]. This identified two independent elements, the *svb[E3N]* and *svb[E6]* enhancers[5,7], which despite having distinct activities during embryogenesis[7] both drive similar expression in adult RNSCs (Supplementary Figure 1c).

Having established that two enhancers drive specific expression of *svb* in the adult stem cells of the renal system, we next assayed consequences of depleting *svb* function in RNSCs. We used a well-controlled genetic system, hereafter referred to as *esg[ts]*. *esg[ts]* ensures RNAi-mediated gene depletion, specifically in stem cells[44], as monitored by GFP expression (Fig. 2a, b and Supplementary Figure 2a). In addition, the expression of *esg[ts]*-driven transgene(s) and/or RNAi is tightly regulated in a temporal manner by a temperature shift, ensuring that stem cell manipulation is turned ON only in adults (3 days after eclosion) to rule out any earlier developmental defects (Fig. 2c). We also developed an image analysis pipeline, allowing automated quantification of the whole population of RNSCs (see Methods). In control conditions, the number of *esg*-positive RNSCs remains stable after adult hatching, with approx. 350 cells *per* tubules (Fig. 2a–c). We only noticed a weak reduction of RNSCs after 1 month. In contrast, *esg[ts]*-driven RNAi depletion of *svb* in adults led to a progressive loss of RNSCs, which were completely absent after 32 days of treatment (Fig. 2a–c). Although not affecting RNSCs before (day 0, Supplementary Figure 2d)—or shortly after (2 days, Fig. 2c)—inducing transgene expression, the effects of *svb* depletion in the adults were already strong following 8 days of treatment, with a two-fold reduction in the number of RNSCs (Fig. 2a–c and Supplementary Figure 2). Similar results were observed when using either a second RNAi construct

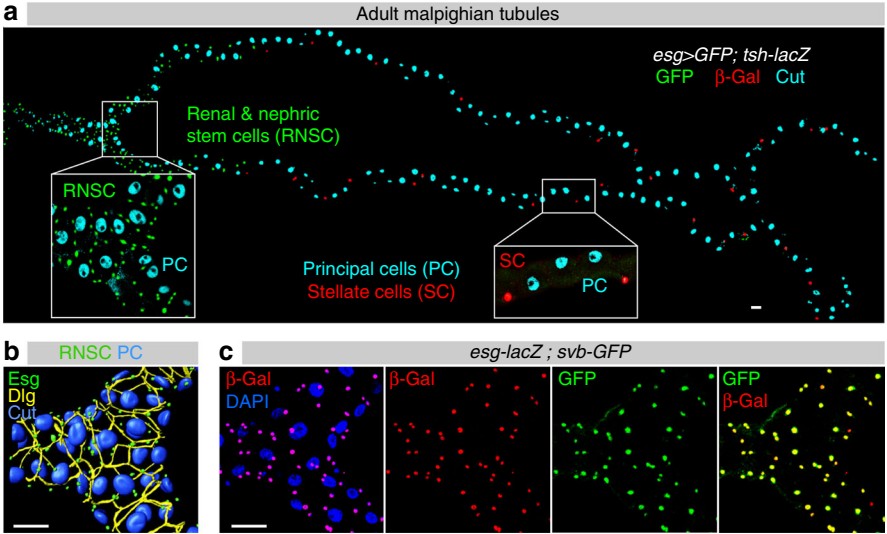

**Fig. 1** *svb* is specifically expressed in renal stem cells. **a** Adult Malpighian tubules are composed of three types of cells. Principal cells (PC) are identified by immunostaining against Cut (cyan) and stellate cells (SC) by *tsh-LacZ* (red). RNSCs, located in the lower tubules, express *esg-Gal4, UAS-GFP* (green). **b** 3D reconstruction of the fork region of Malpighian tubules, with esg-positive RNSCs in green, immunostaining against Discs Large (Dlg) and Cut in yellow and blue, respectively. **c** Expression of *svb* and *esg* as monitored by co-staining for *svb-E-GFP* (green) and *esg-LacZ* (red) enhancers, respectively. Nuclei were counterstained with DAPI (blue). Scale bar is 30 μm

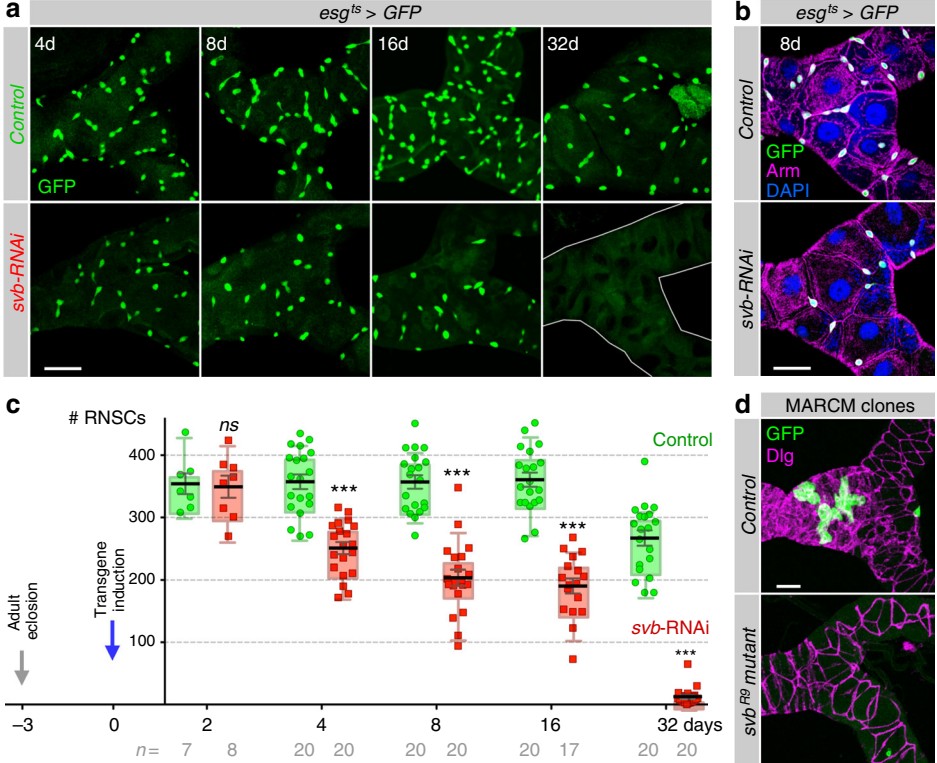

**Fig. 2** svb is required for RNSC maintenance. **a** $esg^{ts}$-driven *svb-RNAi* leads to a progressive decrease of RNSC number compared to controls ($esg^{ts}$ driving only GFP). **b** *svb* depletion eliminates RNSCs, without affecting typical features of remaining stem cells, e.g., *esg* expression and Arm accumulation (see also Supplementary Figure 2). **c** Quantification of the number of RNSCs (*esg*-positive) after 2 ($p = 0.9551$), 4 ($p < 0.0001$), 8 ($p < 0.0001$), 16 ($p < 0.0001$), and 32 days ($p < 0.0001$) of transgene induction in control (green) and *svb-RNAi* (red) conditions. **d** MARCM of control and $svb^{R9}$ clones, positively labeled with GFP (green), 25 days after clone induction. See Appendix for full genotypes. In this and all subsequent figures, each dot corresponds to an independent sample; results were combined from at least two independent experiments. Values are presented as average ± standard error of the mean (SEM) and boxes with whiskers (10–90 percentile). *p*-Values from Mann–Whitney test (ns, $p \geq 0.05$; *$p < 0.05$; **$p < 0.01$; ***$p < 0.001$). Scale bar is 30 μm

(Supplementary Figure 2b), or an independent driver of RNSCs (*dome-MESO-gal4*) to knockdown *svb* (Supplementary Figure 2e). We identified another transcription factor, Hindsight (Hnt), as being specific of RNSCs within Malpighian tubules (Supplementary Figure 2c); and the loss of RNSCs upon *svb* depletion was confirmed by staining against Hnt (Supplementary Figure 2e). Finally, the key role of *svb* in the maintenance of adult RNSCs was further demonstrated by results from genetic mosaics (MARCM[45]), showing that mutant cells bearing strong alleles of *svb* were unable to maintain RNSCs (Fig. 2d and Supplementary Figure 2f, g).

Taken together, these data thus reveal that *svb* is specifically expressed in RNSCs and critically required for the maintenance of the adult stem cell compartment.

**Svb processing is essential for its activity in RNSCs.** In the epidermis, Svb activity relies on a proteolytic processing that causes a switch from a repressor to an activator form[15] (see Fig. 3a). This processing is gated by Pri smORF peptides, which bind to and activate the Ubr3 ubiquitin E3-ligase that, in turn, triggers a limited degradation of Svb operated by the proteasome[18]. Thereby, *pri* mediates a systemic control of Svb maturation since the expression of *pri* is directly regulated by the ecdysone receptor (EcR)[19].

To assess whether the function of Svb in Malpighian tubules also required its proteolytic maturation, we investigated a putative function of *pri* and *ubr3* in RNSCs. We screened a collection of *pri* reporter lines[19,46] and identified two *cis*-regulatory regions driving expression in RNSCs (Fig. 3b and Supplementary Figure 3a, b). Consistently with the expression of *pri* in RNSCs, *pri* knockdown strongly impacted the population of RNSCs (Fig. 3c–e and Supplementary Figure 3c). The effects of RNAi-mediated depletion on RNSCs were even stronger for *pri* than for *svb*, as previously observed in the epidermis[17]. The lack of Pri peptides indeed leads to the accumulation of the Svb repressor[14,17,19],

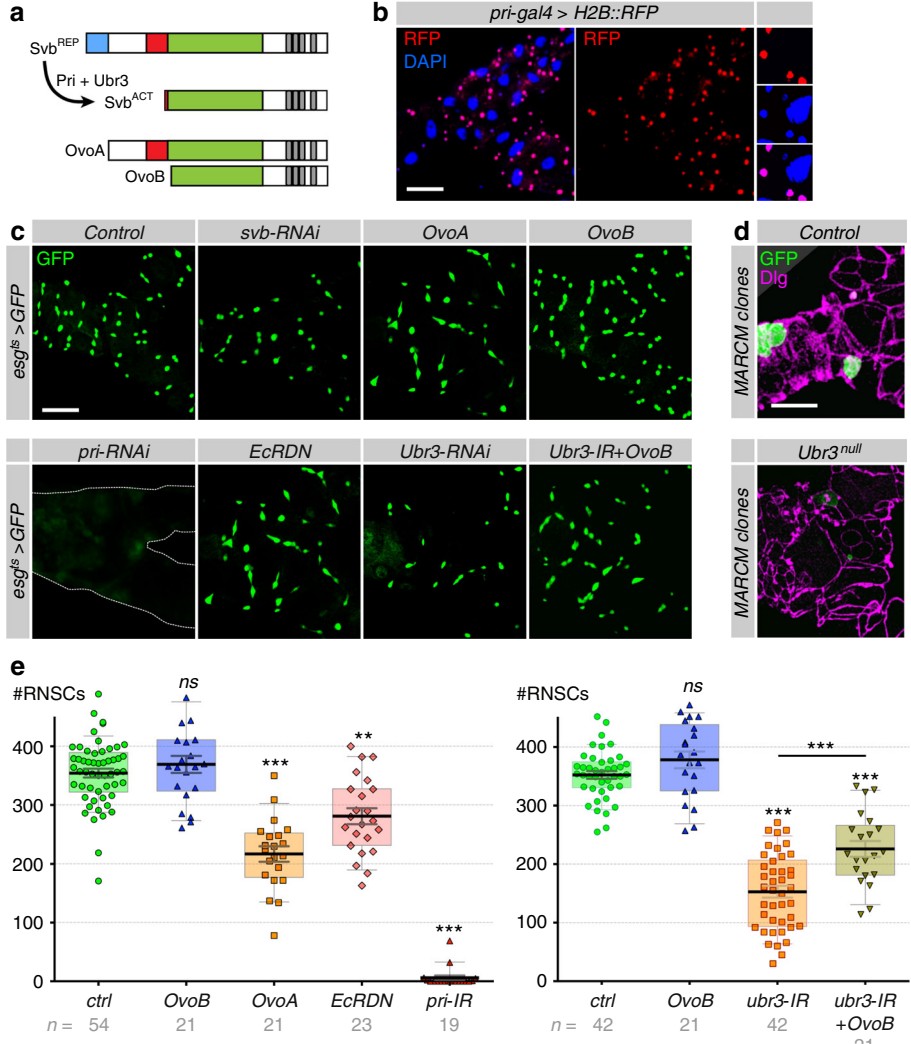

**Fig. 3** Processing of Svb is essential for RNSC maintenance. **a** Schematic representation of Svb maturation, as well as the germinal isoforms OvoA and OvoB that act as constitutive (*pri*-independent) repressor and activator, respectively. **b** Expression of *pri* monitored by the activity of *pri (tal)-Gal4* driving the expression of H2B::RFP (red). Nuclei (DAPI) are in blue. **c** Fork region of Malpighian tubules, with *esg^ts*-driven expression of GFP and the indicated transgenes, after 8 days of induction. **d** MARCM clones (GFP-positive) of control and *ubr3^B* cells, 25 days after induction. **e** Quantification of the number of *esg*+ cells *per* tubule for the indicated genotype. Values are presented as average ± standard error of the mean (SEM) and boxes with whiskers (10–90 percentile). Unless otherwise indicated, *p* values correspond to comparison with control conditions: OvoB ($p = 0.4899$), OvoA ($p < 0.0001$), EcRDN ($p < 0.0001$), *pri*-RNAi ($p < 0.0001$); OvoB ($p = 0.118$), *ubr3*-RNAi ($p < 0.0001$), *ubr3*-RNAi vs *ubr3*-RNAi + *ovoB* ($p < 0.0001$). *p*-Values from Mann–Whitney test (ns, $P > 0.05$; ***$P < 0.001$). Scale bar is 30 μm

which may explain a stronger downregulation of Svb target genes when compared to *svb* mutants[14]. In the case of RNSCs, it is also possible that *pri*-RNAi depletion is more efficient (smORF peptides are likely very unstable when compared to typical protein half-life), and/or that Pri peptides fulfil Svb-independent functions as also reported during embryogenesis[17]. In addition, a dominant negative form of the ecdysone receptor (EcRDN) that abolishes *pri* expression during both embryonic and post-embryonic development[19] was sufficient to reduce the number of stem cells when specifically expressed in adult RNSCs (Fig. 3c–e). Furthermore, we found that *ubr3* was also required for RNSC maintenance, as deduced from results of RNAi-mediated depletion (using three non-overlapping constructs) or genetic nullification[18] of *ubr3* activity (Fig. 3c–e, Supplementary Figure 3d). Finally, the expression of OvoA (see Fig. 3a) that behaves as a constitutive repressor isoform of Svb[17,47,48] mimicked the effects observed in *svb* loss of function conditions (Fig. 3c, e). Reciprocally, the expression of OvoB (see Fig. 3a) that acts as a constitutive activator isoform of Svb[17,47,48] was sufficient to rescue the lack of *ubr3* function (Fig. 3c, e), demonstrating that Svb function in RNSCs relies on its matured transcription activator form.

These results provide compelling evidence that the whole regulatory machinery discovered for its role in the development of epidermal cells[17–19] is also at work in adult RNSCs. We therefore concluded that the post-translational maturation of the Svb transcription factor is essential for the maintenance of RNSCs.

**Svb protects renal nephric stem cells from apoptosis.** The loss of RNSCs observed following the lack of *svb* function or maturation could theoretically result from at least three different causes: (i) lack of proliferation, (ii) precocious differentiation, or (iii) increased cell death. We therefore assayed the putative contribution of each of those aspects in the loss of stem cells resulting from *svb* inactivation.

Consistent with the quiescent behavior of RNSCs, we observed a low frequency of RNSC division in controls, as deduced from staining with the mitotic marker phospho-Histone H3 (Supplementary Figure 4a, b) and as previously noticed[38]. Additional cell lineage analysis using the *esg-Gal4* and *dome-MESO-Gal4* drivers confirmed that adult renal stem cells/progenitors give rise to a progeny of large differentiated cells (Cut positive) in lower tubules (Supplementary Figure 4c, d). Similar analyses using the *Alkaline-Phosphatase 4* driver *(Aph4-Gal4)* that is specific of differentiated principal cells[49] in lower tubules showed no progenitor/progeny figures (Supplementary Figure 4c), supporting that only RNSCs are able to sustain cell renewal. Therefore, even a complete block of stem cell division cannot account for the disappearance of RNSCs observed in the absence of *svb*.

We next investigated a putative influence of *svb* on RNSC differentiation, making use of the lineage-tracing system called ReDDM that has been recently developed for intestinal stem cells[50]. Based on differences in the stability of two fluorescent proteins, ReDDM allows marking renal progenitors that express both mCD8::GFP and H2B::RFP, while their progeny only maintain the very stable H2B::RFP (see Fig. 4a). In control conditions, we detected rare H2B::RFP progeny (Fig. 4a) confirming a low rate of cell renewal in Malpighian tubules[38,39] (Supplementary Figure 4b). Recent work has shown that the expression of *mir-8* (the fly homolog of *mir-200* in vertebrates) downregulates the expression of EMT-inducing factors Escargot and Zfh1 (the homolog of Zeb1), triggering a strong burst of stem cell differentiation in the intestine[50]. Similarly, we found that *mir-8* expression in RNSCs forced *esg+* cells to

differentiate and only rare RNSCs persisted after 8 days of treatment (Fig. 4a, Supplementary Figure 5). Upon *mir-8* expression, the progeny (H2B::RFP-positive, GFP-negative cells) of RNSCs present in lower tubules also expressed Alkaline Phosphatase 4 confirming that the depletion of RNSCs upon *mir-8* overexpression was caused by their premature differentiation (Supplementary Figure 5). In contrast, no significant modification of the progenitors/progeny ratio was observed in *svb*-RNAi conditions when compared to controls, showing that *svb* depletion did not trigger RNSC differentiation (Fig. 4a, Supplementary Figure 5).

Finally, we tested whether *svb*-depleted RNSCs were lost because they underwent apoptosis. As a first step, we assayed consequences of blocking programmed cell death by expressing the viral caspase inhibitor p35[51]. Although the expression of p35 had no detectable effect by itself on RNSCs, it rescued the number of RNSCs when *svb* was depleted (Fig. 4b). Next, we stained for cells undergoing apoptosis using the Terminal deoxynucleotidyl transferase dUTP Nick End Labelling (TUNEL), an assay allowing the detection of apoptotic DNA fragmentation. While apoptotic figures were almost absent in control conditions, expression of the pro-apoptotic gene *reaper* induced a strong increase in the number of TUNEL-positive cells (Fig. 4c). Similarly, knocking down *svb* in stem cells led to frequent apoptotic figures in RNSCs (Fig. 4c) consistently with the progressive decrease in RNSC number observed throughout adult life (Fig. 2c).

Taken together, these data show that the loss of RNSCs observed upon *svb* loss of function is primarily due to stem cell death, indicating that a main role of Svb is to protect adult stem cells from undergoing apoptosis.

**Svb acts downstream of Hippo.** Previous work has shown that the Hippo pathway is a key regulator of the *Drosophila* gut homeostasis, controlling proliferation and survival of stem cells for tissue regeneration[52,53]. Since the Hippo pathway[54,55] is a key sensor of various stresses renowned to induce apoptosis[56,57], we then investigated its function and putative interplay with Svb in the control of RNSC behavior.

The core Hippo complex is composed of two kinases, Hippo (Hpo) and Warts (Wts) and two scaffolding proteins, Salvador and Mob As Tumor Suppressor[54,55]. Activation of Hippo leads to the phosphorylation of the co-transcription factor Yorkie (Yki), preventing its positive action on the transcription of target genes such as *DIAP1* and *bantam*, favoring resistance to apoptosis and proliferation, respectively[54,55]. Consistently, we found that activation of the pathway via Hpo overexpression induced a strong reduction in the number of RNSCs (Fig. 5a, b). Similarly, knockdown of Yki resulted in dramatic RNSC loss (Supplementary Figure 3d and 6a). Conversely, increased Yki activity was sufficient to induce a remarkable increase in the number of RNSCs (Fig. 5), providing additional evidence of the proliferative potential of RNSCs. These data showing the role of the Hippo pathway in the regulation of RNSC survival/proliferation, we next assessed how it interacted with Svb function. Co-expression of OvoB—mimicking a constitutive activator form of Svb (see Fig. 3a)—together with Hpo was sufficient to rescue the loss of RNSCs (Fig. 5). These results therefore suggested that Svb was acting downstream of Hpo. Indeed, the strong increase in the number of renal stem cells observed following increased Yki activity was entirely suppressed upon simultaneous expression either of *svb*-RNAi, or of the constitutive repressor OvoA (Fig. 5). Quantification indicated that *esg+* cells overexpressing Yki were even more sensitive to *svb* loss-of-function than otherwise normal RNSCs (Fig. 5b). Of note, Yki over-proliferating cells showed

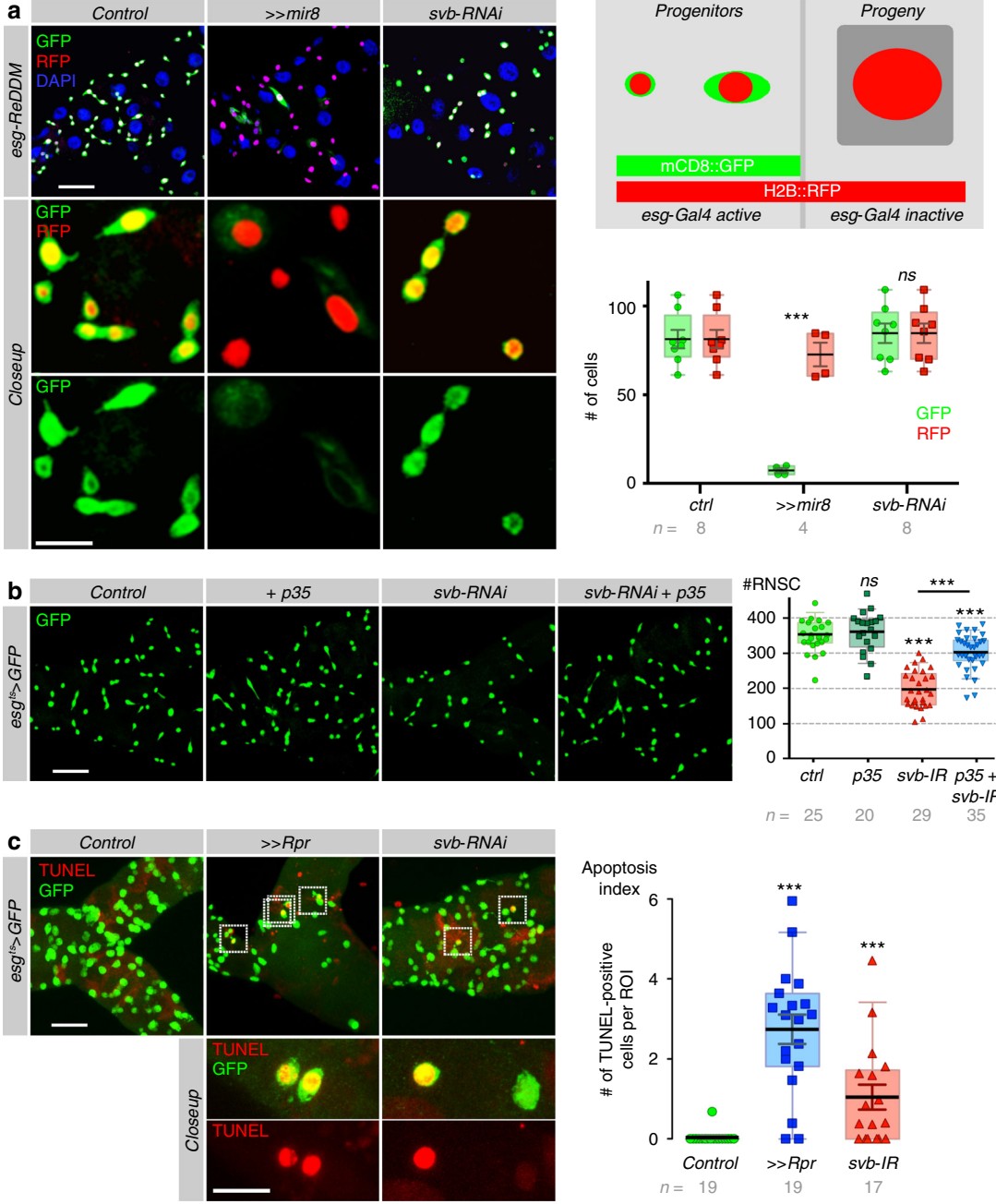

**Fig. 4** *svb* protects RNSCs from apoptosis. **a** Lineage-tracing experiments (*esg*-ReDDM) at 8 days after induction. While stem cells (esg-positive) express both mCD8::GFP (green) and H2B::RFP (red), only the very stable H2B::RFP protein persists in their progeny (*esg* negative). Nuclei are in blue (DAPI). **b** Rescue of *svb*-depleted RNSCs by p35. *esg*^ts^ was used to drive the expression of indicated transgenes (together with GFP), during 8 days. Quantification of esg+ cells is shown at the right. Ctrl vs p35 ($p = 0.6613$), svb-RNAi ($p < 0.0001$), p35 + svb-RNAi ($p = 0.0006$), svb-RNAi vs p35 + svb-RNAi ($p < 0.0001$). **c** TUNEL assays (red) revealing the increase in RNSC apoptosis upon expression of the pro-apoptotic gene *reaper* (*Rpr*, $p < 0.0001$), or RNAi depletion of *svb* ($p = 0.0002$), driven by *esg*^ts^ > GFP (green). Renal stem cells undergoing apoptosis (green cytoplasm and red nuclei) are boxed, and better seen in closeups. The apoptotic index was calculated as the number of TUNEL-positive cells/GFP-positive cells (×100). Values are presented as average ± standard error of the mean (SEM) and boxes with whiskers (10–90 percentile). *p*-Values are calculated using Mann–Whitney test (***$p < 0.001$). $n$ = number of observed tubules. Scale bar is 30 μm, except in closeup panels (10 μm)

aberrant cell morphology reminiscent of tumors derived from intestine stem cells, which also display increased sensitivity to cell death when compared to normal stem cells[58], and that might explain why RNSCS with high Yki levels cannot survive upon Svb knockdown. Hence, the function of Yki in RNSCs requires Svb, suggesting that Svb was interacting with this nuclear effector of the Hippo pathway. Supporting this view, we found that RNSC survival could not be rescued either by expression of OvoB in the

absence of Yki (Supplementary Figure 6a), or by the over-expression of Yki in the absence of Svb (Fig. 5). In contrast, the artificial re-expression of a key target gene of Yki, *DIAP1*, was sufficient to compensate for *svb*-depletion, including in most extreme conditions, i.e., following 32 days of treatment (Figs. 2 and 5a).

In sum, both Svb and Yki are required for RNSC homeostasis, functionally interacting for the survival of adult stem cells. We

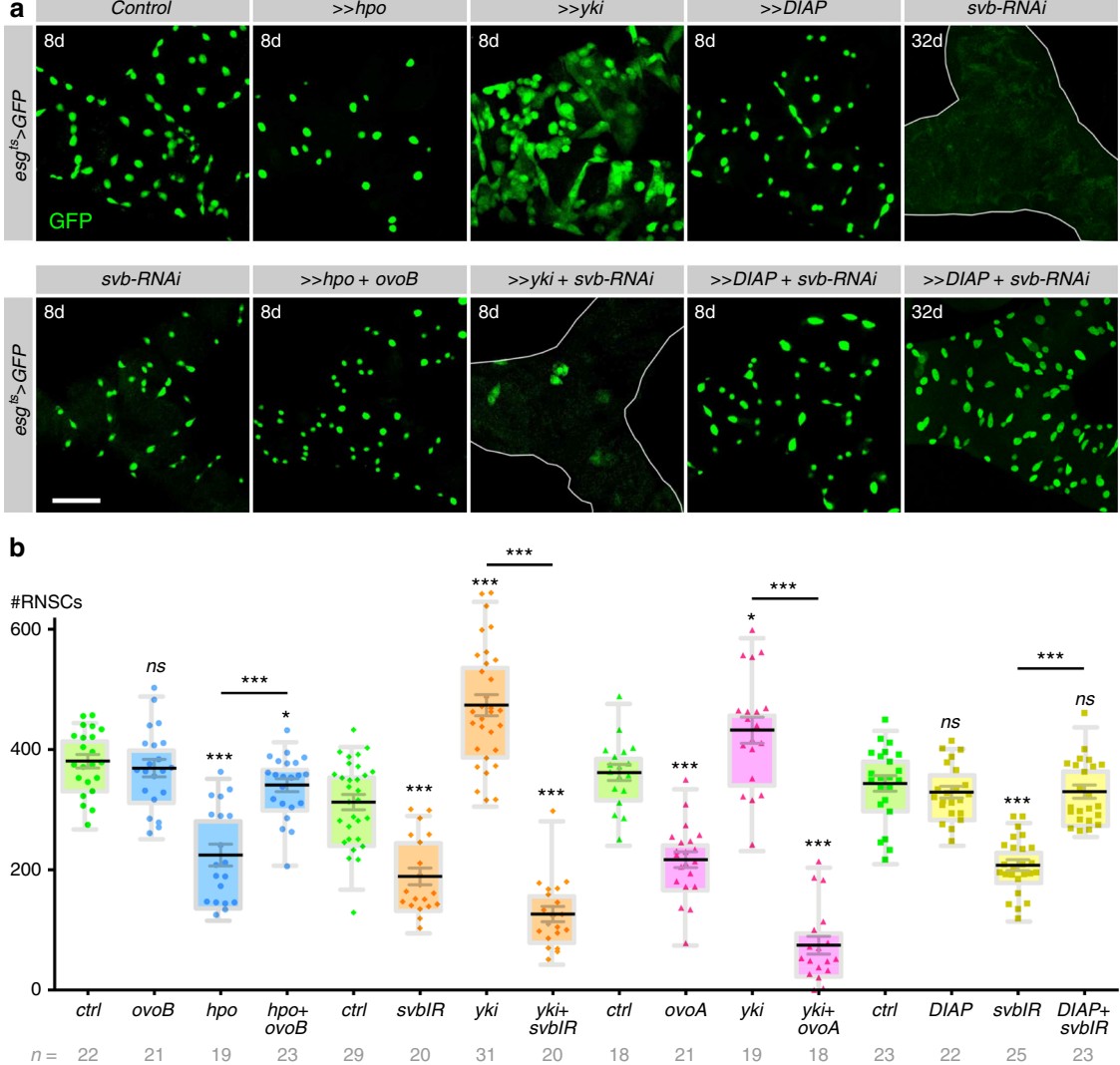

**Fig. 5** *svb* is a member of the Hippo pathway. **a** Pictures of Malpighian tubules with *esg*$^{ts}$-driven expression of GFP (control) and indicated transgenes, at 8 days (8d) or 32 days (32d) after induction. **b** Quantification of *esg*-positive cells in corresponding genotypes. Values are presented as average ± standard error of the mean (SEM) and boxes with whiskers (10–90 percentile). Unless otherwise indicated, *p* values correspond to comparison with control conditions in a same series of experiments, as estimated by Mann–Whitney test (ns, $p \geq 0.05$; *$p < 0.05$; **$p < 0.01$; ***$p < 0.001$). Scale bar is 30 μm

thus concluded that Svb acts downstream of Hippo cytoplasmic core components and, together with Yki, both nuclear factors are required to protect RNSCs from apoptosis.

**Svb as a nuclear effector of the Hippo pathway.** Having established that Svb and Yki functionally interact, we sought to decipher the underlying molecular mechanisms and how these two regulators contribute to the control of relevant effector gene expression, e.g., *DIAP1*.

A first piece of evidence emerged from the comparison of in vivo chromatin immuno-precipitation (ChIP-seq) datasets between Svb[14] and Yki[59]. We found that Svb and Yki share >1.300 common genomic binding sites (Supplementary Figure 7a and Supplementary Table 1) and statistical tests established the significance of this overlap (Supplementary Figure 7b). Interestingly, co-binding of Yki was rare for the direct target genes of Svb identified in the epidermis[12–14], as illustrated by *shavenoid* or *dusky-like* that both lack Yki binding (Supplementary Figure 7c,d). In contrast, Svb was often bound to known Yki target genes, such as *bantam*, *piwi*, *fat*, or *nanos*[60] (Supplementary

Figure 7e, f). Importantly, ChIP-seq also revealed that Svb binds in vivo to an enhancer of *DIAP1* (Fig. 6a), previously identified as a Yki direct binding site[61,62] and that drives specific expression in intestinal stem cells[63]. We therefore tested if Svb might regulate *DIAP1* expression in adult RNSCs. Although weak in control conditions, we observed specific expression of the *DIAP1-LacZ* reporter in RNSCs, which was strongly enhanced upon Yki overexpression (Fig. 6b). This induction of *DIAP1* expression upon Yki overexpression was antagonized by OvoA (Fig. 6b). Similar results were obtained with the isolated *DIAP1-4.3-GFP*[62] enhancer that contains binding sites for Yki and Svb. Although OvoB did not show significant influence (Supplementary Figure 6b), the expression of *DIAP1-4.3-GFP* was again enhanced by Yki overexpression and abrogated upon counteracting Svb activity (Fig. 6b). These results thus strengthen the conclusion that Svb and Yki functionally interact in RNSCs to prevent apoptosis, at least in part through promoting *DIAP1* expression.

The transcriptional activity of Yki/YAP/TAZ is well known to rely on the regulation of its subcellular localization between the cytoplasmic versus nuclear compartments. Assaying a putative

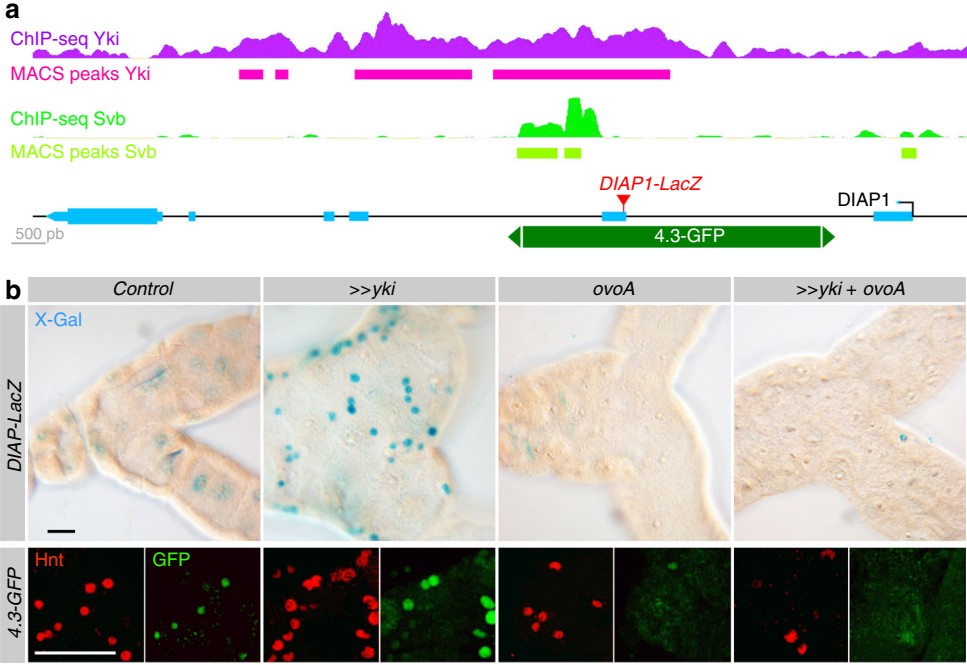

**Fig. 6** Svb interacts with Yki to regulate *DIAP1* expression. **a** Drawing of the *DIAP1* locus. Exons are represented in cyan, the *4.3* enhancer in dark green and the insertion site of the *DIAP1-LacZ* reporter is in red. In vivo ChIP-seq profiles and MACS peaks bound by Svb and Yki are indicated in green and magenta, respectively. **b** *esg^{ts}* was used to drive the expression of *yki*, *svb^{REP}* (*ovoA*), and *yki* together with *svb^{REP}* in RNSCs. The expression of *DIAP1* was followed by the activity of *DIAP1-lacZ* (X-Gal staining, top panel). Expression of the *4.3-GFP* enhancer of *DIAP1* (bottom panel) was followed by immuno-staining against GFP (green) and Hindsight (Hnt, red) revealing the influence of Yki and/or OvoA on RNSCs. Scale bar is 30 μm

influence of Svb on Yki distribution, we did not however detect any consequences of Svb activity (either Svb^{ACT} or Svb^{REP}) on in vivo Yki nuclear accumulation within RNSCs (Supplementary Figure 6c). Yki is unable to bind DNA by itself and need to associate to sequence-specific transcription factors[55]. Hence we assayed whether Yki could associate with Svb to explain the functional interactions observed in RNSCs between these two nuclear factors. Biochemical assays showed that Svb bound to Yki, as seen by reciprocal co-immunoprecipitation in protein extracts prepared from cultured cells (Fig. 7a). Both the full length repressor and matured Svb activator interacted with Yki (Fig. 7a). It has been shown that Yki contains two WW protein domains, mediating interaction with partners bearing PPxY motifs (such as Wts[61] or Mad[64]). Of note, we detected two PPxY motifs within the Svb protein, at position 523 (PPFY) and 881 (PPSY), i.e., within the region common to Svb repressor and activator forms. We found that the mutation of Yki WW motifs was sufficient to abrogate the interaction with both forms of Svb (Fig. 7a). Furthermore, point mutations of Svb PPxY motifs also impaired interaction with Yki, as monitored using immunoprecipitation assays (Fig. 7a). We next investigated the in vivo consequences of impairing Svb/Yki interaction, investigating the functional behavior of the Svb mutant lacking Yki interaction motifs (Svb-PPxY). When expressed in the embryonic epidermis, Svb-PPxY retained the ability to induce ectopic trichomes, consistent with a function of Svb being independent of Yki activity for the terminal differentiation of epidermal tissues (Supplementary Figure 8a). In contrast, Svb-PPxY behaved as a potent dominant negative mutant within adult stem cells. Previous work has shown that Svb^{ACT} and Svb^{REP} display distinct patterns of intra-nuclear distribution, characterized by diffuse nucleoplasmic localization or accumulation in dense foci, respectively[17]. Svb-PPxY::GFP displayed a discrete accumulation in foci, reminiscent of that observed for the repressor (Fig. 7b). Importantly, expression of Svb-PPxY in RNSCs significantly reduced their number, as

observed upon Svb or Yki knockdown (Fig. 7c). These data therefore indicate that Svb physically interacts with Yki (via PPxY and WW motifs, respectively) and impairing specifically this interaction alters adult stem cell behavior.

One important question was whether the interaction between Svb and the Hippo pathway also took place in other tissues. The function of Hippo has been initially described in larval imaginal discs, which give rise to most adult tissues[65] and Yki overexpression promotes cell proliferation in both wing and eye discs[61]. We tested Svb/Yki genetic interactions in the wing using *collier-Gal4* that drives expression in the medial (L3–L4) intervein region. *Yki* expression resulted in the expansion of this region due to tissue overgrowth (Fig. 8a). In contrast, OvoA leads to both a reduction of the L3–L4 region and the absence of epidermal trichomes. As in RNSCs, OvoA was epistatic to Yki, since the wing region expressing both *yki* and *ovoA* was smaller than in controls and lacked trichomes (Fig. 8a). As expected for a mutant unable to interact with Yki, we found that expression of Svb-PPxY did not affect wing cell proliferation. Additional evidence of interaction between the Hippo pathway and Svb/Pri came from genetic assays demonstrating the importance of relative levels of their activity. While lowering *wts* activity causes a wing overgrowth phenotype (Fig. 8b), we noticed that reducing *pri* activity suppressed this phenotype (Fig. 8b, c). Furthermore, *wts* mutation can lead to dramatic defects in wing development when introduced in a given genetic background, and decreasing *pri* dosage was sufficient to almost completely restore normal wings (Supplementary Figure 8b).

In the eye, overexpression of Yki using the *GMR-Gal4* driver promoted extra cell proliferation resulting in an increased eye size. Similar results were obtained following *pri* overexpression, and co-expressing *pri* and *yki* resulted in a synergistic eye growth (Supplementary Figure 9a). Northern blotting of RNAs extracted from adult heads revealed that *DIAP1* mRNA levels were increased following *pri* overexpression (Supplementary Figure 9b), while there was no effect on *yki* or *cycE* mRNA.

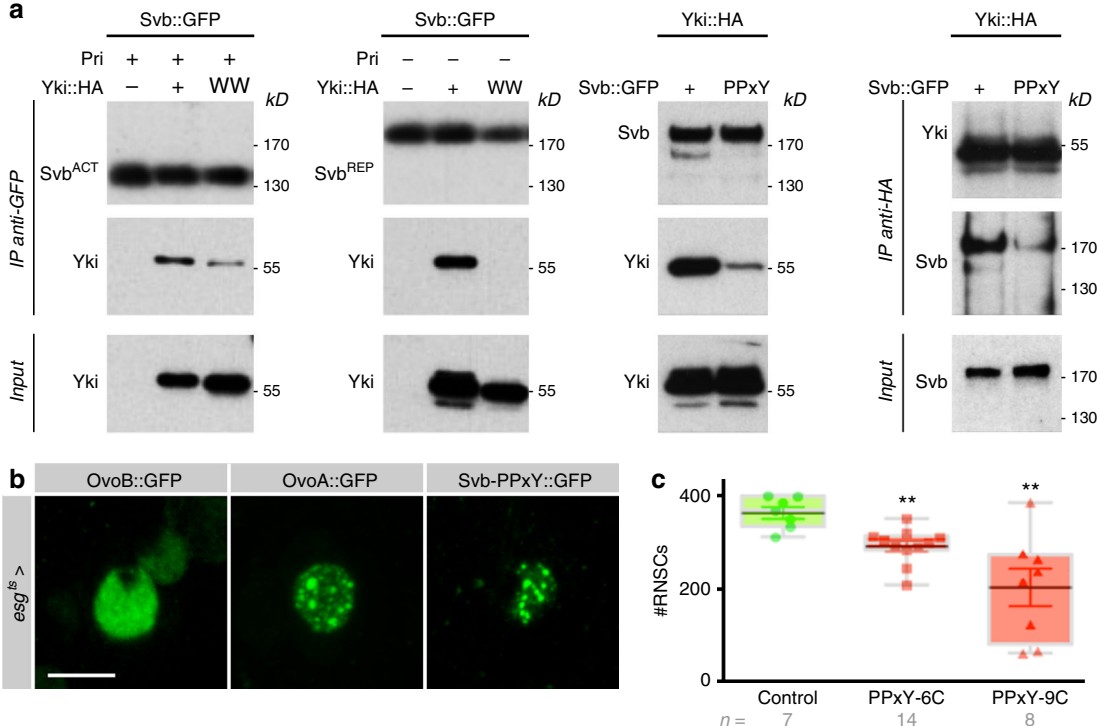

**Fig. 7** The Svb and Yki proteins associate to maintain RNSCs. **a** Co-immuno-precipitation of Svb with Yki. Svb::GFP and Yki::HA were expressed in S2 cells, with or without Pri peptides, and protein extracts were immuno-precipitated using anti-GFP (left panels) or anti-HA antibodies (right panel). Point mutations substituting the WW domains of Yki (YkiWW::HA), or the PPxY motifs of Svb (Svb-PPxY::GFP), prevent Svb/Yki physical interaction. **b** Intra-nuclear distribution of the constitutive activator (OvoB), or repressor (OvoA) forms of Svb within RNSCs. Svb-PPxY displays a pattern of nuclear foci distribution, evoking that of the dominant negative form of Svb (OvoA). **c** Consequences of Svb-PPxY expression in adult RNSCs (driven by $esg^{ts}$-Gal4). Two different genomic insertions allowing the expression of Svb-PPxY::GFP (#6C and #9C) decrease the number of renal stem cells ($p = 0.0059$ and $0.0023$). Values are presented as average ± standard error of the mean (SEM) and boxes with whiskers (10–90 percentile), p-values from Mann–Whitney test (**$p < 0.01$). Scale bar is 3 μm

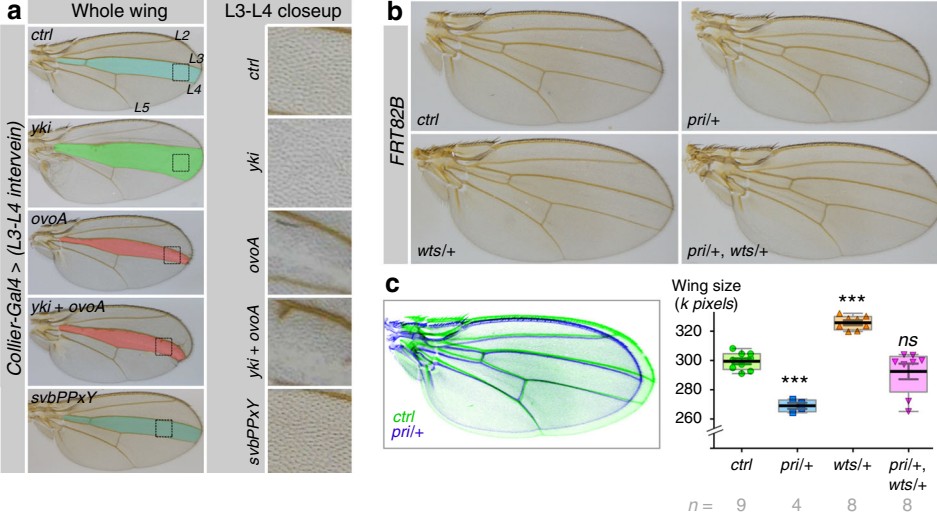

**Fig. 8** Svb/Pri functionally interact with Yki in adult wings. **a** Adult wings expressing the indicated transgenes under the control of col-Gal4, a driver active in the L3-L4 region. **b** Genetic interaction between $pri^{S18.1}$ and $wts^{X1}$ alleles. **c** Overlay between control (green) and $pri-/+$ (blue) wings (left), and quantification of wing area in the different conditions (right). Values are presented as average ± standard error of the mean (SEM) and boxes with whiskers (10–90 percentile). p-Values from Mann–Whitney tests are $pri/+$ (<0.0001), $wts/+$ (<0.0001), $pri/+, wts/+$ (=0.4234) (ns, $p \geq 0.05$; **$p < 0.01$; ***$p < 0.001$).

We interpret these results to imply that Svb functionally interacts with Yorkie, both in adult stem cells and in specific developing tissues, to regulate a subset of transcriptional targets of the Hippo pathway, including the activation of *DIAP1* expression.

## Discussion

Our results show that Shavenbaby is expressed and required for the maintenance of adult renal stem cells in flies, supporting the conclusion that the OvoL/Svb family of transcription factors plays a key and evolutionarily-conserved role in the behavior of progenitors/stem cells.

The role of Svb in adult stem cell maintenance in flies requires both a fine control of its expression and of its transcriptional activity. *Svb* expression in RNSCs involves at least two separable enhancers, driving similar expression patterns. *Svb* was one of the first cases to reveal the functional importance of apparently redundant (or shadow) enhancers in the phenotypic robustness of regulatory networks[5,6] across tissues and development stages[11]. Our data suggest that a similar *cis*-regulatory architecture is also underlying the control of adult stem cells.

RNSCs maintenance further requires a proper post-translational maturation of the Svb protein, in response to Pri smORF peptides. During both embryonic[17] and post-embryonic development[18,19], the main role of Pri peptides is to provide a temporal control of Svb activity, conveying systemic steroid signaling[19]. It is therefore possible that Pri smORF peptides also connect genetic networks to hormonal control for the regulation of adult stem cells. Recent work has shown that various smORF peptides contribute to the regulation of developmental pathways, muscle formation and physiology, etc…[15,66,67], and our findings extend their influence to the control of adult stem cells. It has been proposed that the emerging field of smORF peptides may open innovative therapeutic strategies[16,68], for example peptido-mimetic drugs, which might also be of interest for regenerative medicine.

Our results establish that a main function of Svb in adult stem cells is mediated by a functional interplay with the Hippo pathway, well established for its roles in the control of adult stem cells[52–54]. Our results indicate that Svb behaves as a nuclear effector of Hippo, relying on a direct interaction with Yorkie in order to protect stem cells from apoptosis, at least in part through the regulation of *DIAP1* expression. Analysis of genome-wide binding events further suggests that the Svb/Yki interaction is involved in the control of a broader set of Hippo-regulated genes, including during development. Since both Pri and Ubr3 are also essential for the survival of adult stem cells, it is interesting to note that Ubr3 protects the DIAP1 protein from degradation[69], and direct binding of Ubr3 on the activated form of DIAP1 is elicited in the presence of Pri peptides[18]. Therefore, in addition to the control of *DIAP1* expression (via Svb), Ubr3 and Pri could also stabilize the DIAP1 protein to protect stem cells from apoptosis. Although initially restricted to TEAD transcription factors, the number of Yorkie (YAP/TAZ) nuclear partners is rapidly growing[55]. Interestingly, recent work has shown a direct interaction of YAP/TAZ with the pro-EMT factors Snail/Slug, in the control of stem cell renewal and differentiation[70]. As previously reported for intestinal stem cells[42,43,50], we show that pro-EMT regulators are also required for preventing premature differentiation of renal stem cells. While pro-EMT and OvoL factors are often viewed as antagonistic factors[20,25], in vivo studies in *Drosophila* stem cells show that they both contribute to their maintenance, Svb/Yki preventing their apoptosis and EMT factors their differentiation. Since many studies have implicated the Hippo pathway, pro-EMT and OvoL/Svb factors in various

tumors, new insights into their functional interactions in adult stem cells may provide additional knowledge directly relevant to understand their connections in human cancers.

## Methods

**Drosophila work**. Flies were cultured (unless otherwise noted) at 25 °C, using standard cornmeal food (per liter: 17 g inactivated yeast powder, 80 g corn flour, 9 g agar, 45 g white sugar, and 17 ml of Moldex). Similar results were also observed using a richer medium (same composition except 45 g of yeast powder). Female adult flies were used in all analyses throughout the study and placed on fly food supplemented with fresh yeast, which was changed every 2 days. Conditional expression in RNSCs was achieved by maintaining *tub-Gal80^ts* expressing flies at 18 °C, until adulthood. Eclosed females aged for 3-days were shifted to 29 °C for induction of gene expression and were kept at 29 °C for the indicated period of time (in most cases 8 days). Virgin females bearing *svb^R9*, *svb^PL107*, or *ubr3^B* mutations[18,47] over FM0 balancers were mated with males of the following genotype: *y, w, hs-FLP, tub-Gal80, FRT19A; UAS::mcd8-GFP; tub-Gal4/TM6B, Tb*. Females of the correct phenotype (not *B* and not *Tb*) were heat shocked for 1 h at 37 °C. For G-TRACE experiments, females *w; UAS-RedStinger, UAS-FLP, Ubi-p63E (FRT.STOP)Stinger/CyO* were crossed with *esg-Gal4*, *dome-MESO-Gal4* or *c507-Gal4* males and raised at 18 °C until adulthood. Females were kept at 25 °C and dissected after 20 days. Flies were transferred on fresh food every 2 days and dissected at the indicated time. Detailed information about the genotype of each *Drosophila* stock is given in Supplementary Information.

**Histology**. Tissues were dissected in 1× PBS, fixed in 4% formaldehyde in PBS for 15 min at room temperature, washed for 5 min in PBS containing 0.1% Triton X-100 (PBT) and fixed again during 20 min. Following a 5 min wash in PBT, tissues were blocked for 30 min in PBT containing 1% BSA. Primary antibodies were incubated overnight at 4 °C. Anti-ß-Galactosidase (Cappel Cat#08559761) antibody was used at 1:1000, anti-Cut (DSHB Cat#2B10), anti-GFP (Acris Antibodies Cat#TP401) at 1:200, anti-phospho-Histone H3 (Upstate Cat#TAKMA312B), anti-Disc-large (DSHB Cat#4F3 anti-discs large) at 1:500, anti-Arm (DHSB Cat# N2 7A1) at 1:100, anti-Hnt (DHSB Cat# 1G9) at 1:30. AlexaFluor-488 or -555 secondary antibodies (ThermoFischer Scientific Cat#Z25302 and Cat#Z25205) were incubated for 2 h at room temperature at 1:500. After three washes, tissues were mounted in Vectashield (Vector Laboratories). For X-gal staining, adult tissues were dissected in 1× PBS, fixed in 1% glutaraldehyde in PBS for 15 min at room temperature and washed in PBS. The staining solution was warmed up at 37 °C for 10 min plus another 10 min after addition of 8% X-Gal (5-bromo-4-chloro-3-indoyl-ß-D-Galactopyranoside). The X-Gal solution used to reveal the ß-Galactosidase activity was: 10 mM NaH$_2$PO$_4$·H$_2$O/Na$_2$HPO$_4$·2H$_2$O (pH = 7.2), 150 mM NaCl, 1 mM MgCl$_2$·6H$_2$O, 3.1 mM K4 (FeII(CN)6), 3.1 mM K3 (FeIII (CN)6), 0.3% Triton X-100. Bright-field pictures were acquired using a Nikon Eclipse 90i microscope.

**TUNEL assays**. TUNEL staining was performed following a protocol kindly provided by A. Bergmann. Tissues were dissected in 1× PBS, fixed in 4% formaldehyde in PBS for 30 min at room temperature, washed for 3 times 15 min in PBS containing 0.3% Triton X-100 (PBT) and blocked 15 min in PBT + 5% Normal Goat Serum (PBNT). Anti-GFP antibody was incubated overnight at 4 °C. The following day, tissues were washed with WB (50 mM Tris/HCl pH = 6.8, 150 mM NaCl, 0.5% NP40, 1 mg/ml BSA) and incubated overnight with the secondary antibody. Tubules were then washed 3 times in WB and incubated 30 min at 65 °C with 99 μl of 100 mM Na-Citrate + 1 μl of 10% Triton X-100. Following quick washes with WB, tubules were incubated in 45 μl of labelling solution (in situ cell death detection TMR Red kit, Roche), for 30 min at 37 °C. 5 μl of the enzyme solution was added for 2 h at 37 °C. Tubules were finally washed in PBT and mounted in Vectashield.

**Microscopy, image, and statistical analysis**. Images of whole Malpighian tubules were acquired on a LSM710 confocal scanning microscope (×20 objective), using automated multi-position scan. After stitching, tiled images of individual pairs of tubules were analyzed with IMARIS 8.0 to quantify the number of GFP-positive cells. Data of at least three independent experiments (approx. 20 tubules) were combined. All statistical analyses were carried out using Prism 5 (GraphPad).

Comparisons between normally distributed groups were carried out using Student's *t*-test, unpaired, two-tailed and incorporating Welch's correction to account for unequal variances. One-way ANOVA with Bonferroni correction was used when multiple comparisons were applied. In all figures, ns indicates $p \geq 0.05$, * indicates $0.05 > p \geq 0.01$, ** indicates $0.01 > p \geq 0.001$, and *** indicates $p < 0.001$. Close-up pictures were acquired using Leica SPE and Leica SP8 confocal laser scanning microscopes (×40 and ×63 objectives). Laser intensity and background filtering was set according to the control samples and remained the same for all subsequent samples. The intensity of pictures has been enhanced equally for all images within the same experiment using adjustments in Photoshop CS5. All images were processed using Adobe Photoshop and Illustrator CS5.

**Western blotting and immunoprecipitation**. *Drosophila* S2 cells (Drosophila Genomics Resource Center, stock number: 6) were grown in Schneider medium supplemented with 10% fetal calf serum and 1% penicillin/streptomycin (Invitrogen) at 25 °C. We used either S2 cells or stable cell lines co-expressing the copper-inducible constructs pMT-Svb::GFP and pMT-pri[17]. S2 cells were cultured in six-well plates ($1.75 \times 10^6$ cells/3 ml) and transfected in 100 μl of Opti-MEM (Invitrogen) with 3 μl of FugeneHD (Promega) and the indicated constructs. $CuSO_4$ (final concentration of 1 mM) was used to induce the expression of pMT plasmids. The following plasmids were used: pAc-Yki::HA and its related mutated version pAc-Yki-WW::HA, pAc-Svb::GFP, and pAc-Svb-PPxY::GFP. S2 cells were lysed on ice in 250 μl of ice-cold lysis buffer (150 mM NaCl, 50 mM Tris/HCl pH = 8.0), 0.5% NP40, 1 mM EGTA, and protease inhibitors (Roche). After clearing by centrifugation at 18,000*g* for 10 min, immuno-precipitations from transfected lysates were done in lysis buffer, using anti-GFP antibody (GFP-trap, Chromotek), or anti-HA antibody (Covance) and Fast-Flow Protein G-Sepharose (Sigma #P3296). Immuno-precipitated samples were separated by SDS-PAGE and transferred to PVDF membranes, then blotted by using the following antibodies: anti-GFP (Acris Antibodies, Cat#TP401, 1:10,000), anti-HA (Covance, Cat#BLE901513, 1:2000), and anti-Yki (1:4000, kind gift of K. Irvine). Secondary antibodies were anti-mouse or anti-rabbit IgG-HRP conjugates (Jackson Laboratory, Cat#515-035-003, 1:10,000), detected using the ECL Clarity (Bio-Rad). Uncropped scans of blots are supplied in Supplementary Information.

**Northern blot analysis**. Two days-old adults were frozen with liquid nitrogen and heads were sorted with sieves, followed by RNA purification with Isogen (Nippon Gene). 1 μg of RNA per lane was separated by formaldehyde-agarose gel electrophoresis and then transferred to a nylon membrane (Roche). Hybridization and wash procedures were carried out at 52 and 65 °C, respectively. The filters were reacted with an alkaline phosphatase-conjugated anti-DIG antibody (Roche) and chemiluminescent reactions with CPD-Star (Roche) were detected by LAS 4000mini (GE Healthcare).

## Data availability

The ChIP-seq datasets analyzed during the current study are available in the EMBL-EBI repository: https://www.ebi.ac.uk/biosamples/samples/SAMEA2439952, https://www.ebi.ac.uk/biosamples/samples/SAMN01041482, https://www.ebi.ac.uk/biosamples/samples/SAMN02231205, https://www.ebi.ac.uk/biosamples/samples/SAMN02231213, https://www.ebi.ac.uk/biosamples/samples/SAMN02231206. The data that support the findings of this study are given in Supplementary Table 1. Detailed scripts and computer code used for ChIP-seq analysis are available from the corresponding author on request.

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

## Acknowledgements

We are grateful to FlyBase, Bloomington and Vienna stock centers, Developmental Studies Hybridoma Bank, Drosophila Genomics Resource Center, supported by NIH grant 2P40OD010949, as well as N. Tapon, J. Colombani, K.F. Harvey, D.J. Pan, J. Dow, H. Skaer, M. Crozatier, K. Irvine, A. Bergmann, and M. Dominguez, for providing flies, antibodies, and molecular reagents. We thank B. Ronsin (Toulouse RIO Imaging) for help with microscopy, A. Alsawadi, A. Dib, and M. Soulard for experimental support. We also thank all members of the Payre lab for critical reading of the manuscript. This work was supported by ANR (Chrononet), Fondation pour le Recherche Médicale (Equipe FRM DEQ20170336739), Agence Universitaire de la Francophonie (PCSI AUF-BMO) for D.O. and F.P., PRES IDEX Université de Toulouse for C.P., and by MEXT KAKENHI (JP26113006) for Y.K. J.B. was supported by fellowships from "Ministère de l'Enseignement et de la Recherche" and "La Ligue contre le Cancer". S.A.H. and A.M.-F. were supported by fellowships from "FRM" and "Ligue contre le Cancer", respectively.

## Author contributions

C.P., S.P., Y.K., D.O., and F.P. conceived and directed the project, following initial observations made by D.O. J.B. and C.P. carried out most of the experiments presented here, and other experiments were conducted by S.A.H. (RNAi,), J.Z. (immunoprecipitations), P.V. (molecular biology and transgenics), K.A., Y.Y., and S.I. (adult eyes), and H. C.-D. (genetics). A.M.-F. analyzed NGS data. J.B., C.P., K.A., Y.Y., S.I., H.C.-D., S.P., Y. K., D.O., and F.P. analyzed data and contributed to their interpretation. J.B., C.P., and F. P. prepared the figures and wrote the manuscript. All authors helped to write the paper.

## Additional information

**Competing interests:** The authors declare no competing interests.

