## [Peer Review File · Nature Communications]

Reviewers' comments:

Reviewer #1 (Remarks to the Author):

The manuscript entitled 'Shavenbaby and Yorkie mediate Hippo signaling to protect adult stem cells from apoptosis' by Boheré and colleagues investigates the function of OvoL/Svb in somatic stem cells *in vivo*. While there were several lines of evidence that Svb plays a key role in somatic stem cells (e.g. from studies on cancer as well as somatic stem cells), the mechanism how Svb acts in these non-epithelial cells has remained unclear. Using *Drosophila melanogaster*, and specifically renal and nephric stem cells (RNSCs) as model system for studying Svb's role in somatic stem cells, the authors show that 1) Svb is specifically expressed in RNSCs, 2) that Svb needs to be activated by Pri-mediated post-translational processing, similar as in embryonic epithelial cells, and 3) that Svb's main function in RNSCs is to protect them from apoptosis. By investigating Svb's newly discovered anti-apoptotic activity, the authors furthermore discover a genetic and physical link to the Hippo signaling pathway.

Overall, the paper is well written, the data is of high quality, and the figures are generally well presented (exceptions are commented on below). Given the history of the authors in having discovered the link between Svb and activation by Pri in epithelia, this paper presents a continuation and extension of their previous high-profile mechanistic work on Svb and Pri. The mechanistic insights regarding the link to the Hippo pathway are – to my knowledge – novel, and will make the paper of interest to a more general audience also from the growth/signaling community.

While the data overall is of high quality and clear, the two key findings of this manuscript (1) Svb's mode of action as an anti-apoptotic factor, and 2) Svb's connection to the Hippo pathway) will need to be presented clearer since they leave room for doubts/open questions in the reader that are not addressed. In the following I provide specific comments and suggestions to both points.

1) Svb's anti-apoptotic activity in RNSCs

This part would become much more solid by two really showing that RNSCs undergo apoptosis in the absence of active Svb. The current data is based on a rescue by inhibiting apoptosis, yet as a reader I would want to see data that shows that there are indeed more apoptotic cells. The authors comment that detecting apoptosis is difficult, yet it is unclear whether this has been tried at all and which methods have been tried. I would suggest for example a standard TUNEL assay, or annexin5-GFP, or detecting activated Caspase in those cells.

This is even more important since the data shown in Figure 3a, that is supposed to demonstrate no effect on differentiation, does in my opinion not allow drawing this conclusion. I am not sure what exactly was quantified in the lower panels, yet I can detect some red only nuclei (2) and 2 weak green in the upper large picture from sv5-RNAi, while no red only cells are visible in the control situation. The quantification underneath for the svb-RNAi does not reflect this difference, yet there does seem to be one, which would suggest that Svb might also have a role in keeping the cells from differentiating, which might contribute to the loss of RNSCs in svb-RNAi in addition to apoptosis. How many cells/images/flies were used for the quantification? This quantification would need to be repeated and done properly (with statistics, see below).

2) Svb-Hippo pathway link

The biochemical (Svb-Yorkie) and genetic link of Svb to the Hippo pathway is convincing, yet the positioning of Svb within the Hippo pathway is hard to follow, particularly since some of the data presented seem contradictory. Proposed model from the authors: Svb and Yki act together to induce expression of DIAP1, which block pro-apoptotic pathway. Hippo can block Yki and Svb, leading to the induction of apoptosis. This does not fit with the original hypothesis of cell shape changes, that are caused by svb depletion, inducing apoptosis. It is also unclear whether Svb and Yorkie act in parallel,

or – as suggested elsewhere – whether their anti-apoptotic activity mutually depend on each other? The fact that the apoptosis upon Hippo overexpression (Fig 4a) can be rescued by ovoB overexpression seems to suggest that the anti-apoptotic activity of ovoB/Svb does not depend on Yorkie (which should be degraded upon Hippo OE)? This would need to be worked out better and/or explained better. A model would be very helpful that places Svb in the Hippo pathway.

What also remains entirely unclear is why Yorkie overexpression + svb-RNAi causes a more severe apoptosis than svb-RNAi on its own. How can this be explained?

The authors favor the idea that Svb might 'bring' Yorkie to the right places on the DNA – yet the overlap between the genomic peaks is very little (~10% only) – how would that targeting then work? An alternative hypothesis could be that Svb brings Yorkie into the nucleus – this could be checked in the different mutants (svb-/-, OvoA and OvoB OE).

Apart from these two main critiques, I have listed further suggestions for improvements of the paper below that should be addressed (in the order as they appear in the paper):

- In general, all quantifications need to have significance values (p-values; error bars in 3a; give the numbers of flies/cells counted/assessed)
- Figure 1b: merge svb-GFP and b-Gal images and show the 'raw' data, not something with shading (?)
- Figure 1c: what is GFP
- Figure 1d: t = 0 would need to be shown as well – are the numbers of cells at the beginning the same? This would be important to know, since otherwise Svb might also have a role in RNSC establishment and not only maintenance; also please explain the *esg[ts]* system to readers not familiar with this system
- Figure 1e: also show a picture of an earlier time-point; if maintenance affected, then it would be important to know how big the initial clone-size was!
- Figure 2: why is there such a dramatic difference between priRNAi (no cells left already at 8 days after induction) and svb-RNAi? Can this be explained by more efficient RNAi, or the fact that in priRNAi the repressor function of svb is still left (which would potentially suggest that Svb-repressor acts pro-apoptotic?). Or does Pri have other targets in addition to Svb? To address this, it will be crucial to test whether ovoB expression can also rescue pri-RNAi (as it is shown in Figure 2d for *ubr3*-RNAi).
- The cell shape changes in svb-RNAi cells (Fig 3b) do not fit in this order, and do not seem to fit in general. As a link between Svb and Hippo, this is not convincing. A direct link Svb → Apoptosis → Hippo, would make more sense. Not sure if this experiment adds anything, and only contradicts the final conclusion of Svb acting downstream of Hippo.
- Molecular interaction between Yorkie and Svb: do the reverse IP (co-IP of Yorkie and check which isoform of Svb (the repressor or activator) comes down – this seems important to know!
- Figure 4b (genomic peaks): raw data needs to be shown here somewhere (read accumulation over genomic regions) – at least in supplement – to judge how the peaks were called.
- Induction of DIAP1 expression should also be tested with svb/OvoB (Fig. 4d)

Reviewer #2 (Remarks to the Author):

In this manuscript, Bohère et al describe a novel role for the Shavenbaby/Ovo (Svb) transcription factor in maintenance of renal/nephric stem cells (RNSCs) in the *Drosophila* malpighian tubules, which are functionally analogous to the mammalian kidney. These cells are derived from the same lineage that populate the gut intestinal stem cells (ISCs) niche but have not been studied very thoroughly. This study presents evidence that Svb interacts physically and functionally with the Yorkie co-activator protein to control expression of one of its classic targets, Diap1, which in turn is required to block

apoptosis of RNSCs. The Polished rice (Pri) peptides are required for Svb processing to its active form, and in accordance with the model, Pri is also required for RNSC maintenance and seems to functionally interact with Yki. The paper also offers some evidence that Svb also supports Yki-dependent gene expression in two more classic epithelial models, the eye and wing disc, suggesting that the manuscript has hit upon a generalizable mechanism.

Overall, the manuscript has some elements of a compelling story, including the clean biochemical evidence of a PPxY-based interaction between Yki and Svb, a consistent pattern of genetic interactions between Hippo and Svb/Pri alleles in control of RNSCs numbers, and in vivo reporter data that support a role for Svb in control of diap1 transcription. However it also has some gaps that when filled may provide more compelling insight into what's going on at the intersection of the Svb and Hippo pathways.

Main comments:

1) The Svb:Yki interaction is a key advance provided by the paper, but the biochemistry has no in vivo proof of relevance. Does a version of Svb with mutations engineered into the two PPxY sites act like a dominant negative with respect to RNSC number or control of Diap1?

2) Does Yki ChIP onto the diap-1 promoter depend on Svb?

3) Throughout the paper, single UAS-RNAi lines are used to assess gene function. It is necessary to show that a second RNAi line, or preferably a genomic allele, recapitulates the phenotype. The authors use genomic alleles in MARCM, but its not done in a way that tests function in RNSCs. e.g. in Fig 1, the authors use tub-Gal4 MARCM to show that svb mutant clones survive poorly in the malphigian tubule, but there's no way to know that the marked cells are RNSCs or one of the other cell types in the organ. As is the data in Fig. 1e simply say that Svb is required for clonal survival.

4) There is variability in the length of RNAi knockdown in different experiments that raises some questions. For example, the Pri RNAi phenotype is quite a bit stronger than the Svb RNAi phenotype: Pri kd eliminates RNSCs in 8 days but Svb RNAi takes 32 days to eliminate them. This could be a trivial consequence of kd efficiency, or it could argue that the Pri peptides have roles beyond Svb, which could profoundly affect how data is interpreted throughout the paper. Day8 Svb kd is used as the basis for modification/rescue by Yki. Does this look the same at 32 day kd when the Svb phenotype is stronger?

5) In a related concern, the rescue of svb-RNAi by UAS-p35 in Fig3c is important as it justifies the leap to apoptosis as the cause of RNSC loss. However, its done with 8day svb RNAi which is a much more subtle phenotype than the 32day RNAi shown in Fig 1. I would be much more convinced if p35 rescued the RNSC loss after 32 days of svb RNAi.

6) The data implying a role for Svb in Yki phenotypes in eyes and discs is exciting, as it implies a potentially conserved relationship across tissues. However, the disc data is incomplete and inserted into supplemental data section. It should be strengthened and moved into the main body of the text:

e.g. Figs S3D and E respectively show the effects of OvoA or Pri co-expression with Yki in the wing or eye.

The images in S3D nicely show OvoA shrinks Yki-driven wing growth; however, this would be much stronger if they could show this correlates with suppression of Yki driven expression of one of its targets.

In the eye data in S3E, GMR-Pri has a hyperplastic phenotype on its own, but the combined GMR-yki/Pri eye seems to change shape more than size (flatter and more hammer-like). Organ shape is an overly crude way to measure effects on a putative shared Svb/Yki transcriptome. The authors should simply test whether coexpression of Pri or Svb enhances Yki-driven diap-lacZ (or ex-lacZ or bantam-lacZ) and reciprocally whether RNAi of Svb or Pri suppresses this effect.

Other comments:

1) Caspase-driven death is fairly rapid, but that latency to death following kd of Svb is on the order of days. Why is this? If Yki and Svb control Diap1 transcription, which interacts with and inhibits primed caspases, they should die fairly quickly. On a related note, overexpressed Rpr is not sufficient to kill ISCs (Jin et al, 2017, Stem Cell Reports; from the Edgar lab), so I'm a little concerned that the effect of Diap in RNSC survival may not be via the classic Rrp/Hid-Diap1-Caspase pathway.

2) In Figures 2, 3 and 4 the percent of surviving RNSCs is presented in a series of individual green bar graphs to the right of an optical slice. These quantitative data are critical and claim to show rescue and epistasis between pri, hpo, yki and svb, but they are hard to evaluate since they are not combined into one graph with stats & p-values. They should all be presented as in Fig. S3, panel F...why is this key data in supplement?

7) The data in Fig. 3b on cell size data not meaningful vis a vis apoptosis or Hippo status. Apoptotic cells bleb and fragment. Are svb RNAi cells positive for tunel?

Reviewer #3 (Remarks to the Author):

In this interesting manuscript, Bohere and colleagues present strong evidence that the transcription factor Shavenbaby (Svb) is expressed in the adult Renal/Nephritic Stem Cells (RNSCs) of *Drosophila melanogaster*, and it is critically required for their survival. More specifically, they relate Svb with the Hippo pathway, disclosing its physical interaction with Yorkie (Yki) which, together, positively regulate the expression of the apoptosis inhibitor DIAP1. In addition, they demonstrate that the processing of Svb by pri and ubr3 is required for its correct function, in a similar way to other tissues already reported in the literature.

They show that loss of function of Svb (either with RNAi or the constitutive repressor OvoA) induces a serious loss in the population of RNSCs in the ureters and lower part of the Malpighian tubules.

Furthermore, they determine the interaction of Svb and Yki, either by co-immuno-precipitation and epistatic experiments, demonstrating that their genetic and physical interaction is required for the correct expression of DIAP1, and therefore, inhibiting apoptosis in the tubules. The experimental data presented is elegantly convincing, and the conclusions fit in to the question initially asked.

Main points:

Are these really stem cells? Since Hou's paper (which may have issues), there has been a lack of evidence that these cells actually divide, and I notice that the authors had trouble finding evidence of mitosis in this MS. Alternatively, they have been described as myoendocrine by Garaoya and by Sozen, and as 'tiny cells' by the latter. I don't think this detracts from the publishability of this MS; but I think that, to prevent future embarrassment, the authors should at least put a sentence in the introduction acknowledging that their identity is not entirely settled. Better still, if they have evidence that these cells are mitotically competent, they should include it in this MS.

The authors identify the RNSCs by the GFP positive cells under the *esgGal4* driver, and they present a

new protein expressed in these cells, Hindsight, as specific of these stem cells. Although it is interesting that this protein is specifically expressed in the RNSCs, it is not clear the relation that this protein has with them. In addition, the loss of GFP due to loss of RNSCs in some of the figures doesn't make clear the point that these cells are actually not there. Maybe it would be good to include a nuclear counterstaining (DAPI/To-Pro3) in, at least one figure, to show this (e.g. Fig1.c), as well as more information about Hindsight, or other stainings that show the identity of RNSCs (maybe staining for Delta, or another stem cell marker, to give strength to the fact that these are RNSCs).

Minor points:

In the figure 4a, the authors show that ovoB is able to rescue the loss of RNSCs caused by the overexpression of hpo (which, in theory, it will induce the degradation of more yki, so it can be considered similar to yki-RNAi). However, this result doesn't seem to be in line with the phenotype shown in Suppl. Figure 3c, in which ovoB is not able to rescue yki-RNAi phenotype. It is true that the phenotype of the overexpression of hpo (Figure 4a) is milder than the yki-RNAi (Suppl. Figure 3c), and the phosphorylation of yki depends of other components of the Hippo pathway, but it is not clear the relation between these two apparently contradictory results.

In the figure 4d and 4e, it is beautifully shown how DIAP1 is induced by svb/yki, and that the constitutive repressor OvoA is enough to block the expression of DIAP1 even when yki is overexpressed. Have you tried to do the opposite experiment? Observe the expression of DIAP1 in a yki-RNAi/OvoB background? In addition, it would be informative another experiment to show ongoing apoptosis of RNSCs (Caspase staining/western-blot, or TUNEL assay?), if possible.

Finally, in the overall manuscript it shows that the architecture of the tubule (at least the ureter) looks normal, even when the RNSC population is absolutely abolished (e.g. in Figure 2d). It looks surprising that the tubule – and the organism- survives without major defects even without these (stem?) cells. Have you observed any defects in later stages, both morphologically or physiologically? (secretion rates, stress tolerance, or response to tissue damage?). In my opinion, this would strength the requirement for the RNSCs and their function in the adult tubule.

Reviewer #1

Overall, the paper is well written, the data is of high quality, and the figures are generally well presented (exceptions are commented on below). Given the history of the authors in having discovered the link between Svb and activation by Pri in epithelia, this paper presents a continuation and extension of their previous high-profile mechanistic work on Svb and Pri. The mechanistic insights regarding the link to the Hippo pathway are – to my knowledge – novel, and will make the paper of interest to a more general audience also from the growth/signaling community.

While the data overall is of high quality and clear, the two key findings of this manuscript (1) Svb's mode of action as an anti-apoptotic factor, and 2) Svb's connection to the Hippo pathway) will need to be presented clearer since they leave room for doubts/open questions in the reader that are not addressed. In the following I provide specific comments and suggestions to both points.

1) Svb's anti-apoptotic activity in RNSCs

This part would become much more solid by two really showing that RNSCs undergo apoptosis in the absence of active Svb. The current data is based on a rescue by inhibiting apoptosis, yet as a reader I would want to see data that shows that there are indeed more apoptotic cells. The authors comment that detecting apoptosis is difficult, yet it is unclear whether this has been tried at all and which methods have been tried. I would suggest for example a standard TUNEL assay, or annexin5-GFP, or detecting activated Caspase in those cells.

As suggested by the reviewer, we performed TUNEL assays in wild type and *svb*-RNAi conditions, as well as upon overexpression of the proapoptotic gene *reaper* as a positive control. These results further demonstrate that the lack of Svb in adult stem cells induces apoptosis and they are presented in the new **Figure 3c**.

This is even more important since the data shown in Figure 3a, that is supposed to demonstrate no effect on differentiation, does in my opinion not allow drawing this conclusion. I am not sure what exactly was quantified in the lower panels, yet I can detect some red only nuclei (2) and 2 weak green in the upper large picture from *svb*-RNAi, while no red only cells are visible in the control situation. The quantification underneath for the *svb*-RNAi does not reflect this difference, yet there does seem to be one, which would suggest that Svb might also have a role in keeping the cells from differentiating, which might contribute to the loss of RNSCs in *svb*-RNAi in addition to apoptosis. How many cells/images/flies were used for the quantification? This quantification would need to be repeated and done properly (with statistics, see below).

It is sometime difficult to estimate signal intensity in overlay pictures and we developed an automated pipeline (ImageJ) for ReDDM analysis using the same setup and thresholding for all images. We now provide larger views of green/red pictures (**Fig. 3a**), as well as separate channels in supplementary info (**Fig. S4 d**).

We yet agree that the former plot was not optimal and now switch to a clearer representation, showing each point (corresponding to independent samples). Following the reviewer recommendation, in this and all other figures; we also provide detailed

information in the number of samples and statistical tests. In general, our data correspond to results from three independent experiments, combining ≥ 20 samples.

2) Svb-Hippo pathway link

The biochemical (Svb-Yorkie) and genetic link of Svb to the Hippo pathway is convincing, yet the positioning of Svb within the Hippo pathway is hard to follow, particularly since some of the data presented seem contradictory. Proposed model from the authors: Svb and Yki act together to induce expression of DIAP1, which block pro-apoptotic pathway. Hippo can block Yki and Svb, leading to the induction of apoptosis. This does not fit with the original hypothesis of cell shape changes, that are caused by *svb* depletion, inducing apoptosis. It is also unclear whether Svb and Yorkie act in parallel, or – as suggested elsewhere – whether their anti-apoptotic activity mutually depend on each other? The fact that the apoptosis upon Hippo overexpression (Fig 4a) can be rescued by OvoB overexpression seems to suggest that the anti-apoptotic activity of OvoB/Svb does not depend on Yorkie (which should be degraded upon Hippo OE)? This would need to be worked out better and/or explained better. A model would be very helpful that places Svb in the Hippo pathway.

We acknowledge that the description of Svb/Pri versus the Hippo pathway was somehow confusing and extensively modify the text according to the suggestions of reviewers. Importantly, new data (*in vitro*, *in vivo*, dosage sensitive genetic assays) should help clarifying the relationships between Svb and Yki (**Figure 5, Sup 5, Sup 7**).

What also remains entirely unclear is why Yorkie overexpression + *svb*-RNAi causes a more severe apoptosis than *svb*-RNAi on its own. How can this be explained?

Recent work from Norbert Perrimon's lab has shown that tumorigenic cells, that is the case of Yki overexpressing cells are more sensitive to cell death (Ma *et al.*, Dev Biol 2016). We believe that is the reason why the overexpression + *svb*-RNAi causes a more severe apoptosis than *svb*-RNAi on its own.

The authors favor the idea that Svb might 'bring' Yorkie to the right places on the DNA – yet the overlap between the genomic peaks is very little (~10% only) – how would that targeting then work? An alternative hypothesis could be that Svb brings Yorkie into the nucleus – this could be checked in the different mutants (*svb*^{-/-}, *OvoA* and *OvoB* OE).

As suggested, we addressed this possibility and look at the subcellular localization of Yki in *svb*-RNAi, *OvoA* and *OvoB* contexts. The results are presented in **Figure Sup. 5c** and they do not support that Svb plays a major role in regulation of Yki nuclear localization. In contrast, the interaction with Yki may be important for the intra nuclear distribution of Svb in RNSCs as the PPxY-mutant behaves as *OvoA* (**Figure 5d,e**).

Apart from these two main critiques, I have listed further suggestions for improvements of the paper below that should be addressed (in the order as they appear in the paper):

- In general, all quantifications need to have significance values (p-values; error bars in 3a; give the numbers of flies/cells counted/assessed)

The number of samples, error bars and p values have been added to each figure (**Figure 1, 2, 3, 4, 5**) and supplemental material (**Figure Sup. 2,3,4,7**).

- **Figure 1b:** merge svb-GFP and b-Gal images and show the 'raw' data, not something with shading (?)

Raw data of merged picture is now presented in **Figure 1b**.

- **Figure 1c:** what is GFP

GFP was driven by *esg^{ts}-Gal4*, as now better presented in **Figure 1c,c'**.

- **Figure 1d:** t = 0 would need to be shown as well – are the numbers of cells at the beginning the same? This would be important to know, since otherwise Svb might also have a role in RNSC establishment and not only maintenance; also please explain the *esg^{ts}* system to readers not familiar with this system

We show that the RNSC compartment is unaffected before (day 0, **Figure Sup. 2d**), or shortly after (2 days, **Figure 1d**) induction of *svb*-RNAi expression. Both Figure 1 and the text (**P6**) have been modified to better explain how the *esg^{ts}* system works.

- **Figure 1e:** also show a picture of an earlier time-point; if maintenance affected, then it would be important to know how big the initial clone-size was!

Unfortunately, MARCM mosaics prevent imaging the same clones at several times of adult life. We add new clonal analyses (**Figure Sup. 2f,g**), including high resolution pictures and anti-Cut staining, showing that *esg⁺* RNSCs are unable to proliferate and/or differentiate in the absence of *svb* function. We also add new data in **Figure 1c'** and **Sup. 2a** (co-staining with Armadillo and nuclei) further demonstrating the disappearance of stem cells upon *svb* knockdown.

- **Figure 2:** why is there such a dramatic difference between *pri*-RNAi (no cells left already at 8 days after induction) and *svb*-RNAi? Can this be explained by more efficient RNAi, or the fact that in *pri*-RNAi the repressor function of *svb* is still left (which would potentially suggest that Svb-repressor acts pro-apoptotic?). Or does Pri have other targets in addition to Svb? To address this, it will be crucial to test whether *ovoB* expression can also rescue *pri*-RNAi (as it is shown in Figure 2d for *ubr3*-RNAi).

The reviewer is right and the different hypotheses are not mutually exclusive. Indeed, the expression of Svb^{REP} is sufficient to reduce the number of RNSCs (Figure 2). *pri* mutant phenotypes are also stronger than the lack of *svb* in the epidermis (Kondo et al, Science 2010), including for the down-regulation of Svb direct target genes (Menoret et al, Genome Biol 2013). Like other smORF peptides, Pri are expected to be extremely

short lived, which may improve the efficiency of RNAi-mediated extinction of *pri* when compared to “regular” large-sized proteins. For example, it has been shown that the maternal load of Ubr3 protein can compensate for the lack of zygotic expression, throughout the whole embryogenesis (Zanet *et al.* Science 2015). Finally, there are additional functions of Pri peptides (*i.e.*, *svb* independent) and Ubr3 contributes to protect against apoptosis (Ref). These different aspects that may explain the relative severity of *pri* phenotype are now clearly presented in the text (P7).

- The cell shape changes in *svb*-RNAi cells (Fig 3b) do not fit in this order, and do not seem to fit in general. As a link between Svb and Hippo, this is not convincing. A direct link Svb → Apoptosis → Hippo, would make more sense. Not sure if this experiment adds anything, and only contradicts the final conclusion of Svb acting downstream of Hippo.

We agree with the reviewer and remove this data in the revised version.

- Molecular interaction between Yorkie and Svb: do the reverse IP (co-IP of Yorkie and check which isoform of Svb (the repressor or activator) comes down – this seems important to know!

We performed these additional co-IPs, which confirm the interaction of Yki with both Svb^{REP} and Svb^{ACT} (Figure 5c). Also, as suggested by Rev#2, we provide evidence that point mutations of either WW-domains in Yki, or PPxY-motifs in Svb, are sufficient to disrupt the interaction (Figure 5c).

- **Figure 4b** (genomic peaks): raw data needs to be shown here somewhere (read accumulation over genomic regions) – at least in supplement – to judge how the peaks were called.

ChIP-seq data and Yki-dependent activity of the DIAP1 (4.3) enhancer have been published previously (Zhang *et al.*, Dev Cell 2008). To ensure faithful comparison of Yki and Svb datasets, we however entirely reprocessed both raw data (fastq reads) using the same pipelines, giving very similar peak calling to those previously published. As requested, we now present ChIP-seq profiles for both Yki and Svb binding (Figure 5a).

- Induction of DIAP1 expression should also be tested with *svb/OvoB* (Fig. 4d)
We performed this experiment showing that OvoB is not sufficient to significantly increase DIAP-GFP expression (Figure Sup. 5b).

Reviewer #2

Overall, the manuscript has some elements of a compelling story, including the clean biochemical evidence of a PPxY-based interaction between Yki and Svb, a consistent pattern of genetic interactions between Hippo and Svb/Pri alleles in control of RNSCs numbers, and *in vivo* reporter data that support a role for Svb in control of diap1 transcription. However, it also has some gaps that when filled may provide more compelling insight into what's going on at the intersection of the Svb and Hippo pathways.

Main comments:

1) The Svb:Yki interaction is a key advance provided by the paper, but the biochemistry has no *in vivo* proof of relevance. Does a version of Svb with mutations engineered into the two PPxY sites act like a dominant negative with respect to RNSC number or control of Diap1?

We engineered a mutant form of Svb bearing point mutations of the PPxY-motifs and assayed consequences on Svb activity. *In vitro* assays show that (as observed for the mutation of Yki WW-domains) point mutations of Svb-PPxY motifs abrogate the biochemical association between Yki and Svb (**Figure 5c**). Furthermore, and as anticipated by the reviewer, expression of the Svb-PPxY mutant decreases the number of RNSCs (**Figure 5e**).

2) Does Yki ChIP onto the diap-1 promoter depend on Svb?

We add new data showing that the subcellular localization of Yki is not dependent of Svb (**Figure Sup. 5**). As deduced from the *in vitro* and *in vivo* behavior (of the PPxYmut version of Svb, it seems that Yki might in turn contribute to control the localization and/or activity of Svb.

3) Throughout the paper, single UAS-RNAi lines are used to assess gene function. It is necessary to show that a second RNAi line, or preferably a genomic allele, recapitulates the phenotype. The authors use genomic alleles in MARCM, but its not done in a way that tests function in RNSCs. e.g. in Fig 1, the authors use tub-Gal4 MARCM to show that *svb* mutant clones survive poorly in the malphigian tubule, but there's no way to know that the marked cells are RNSCs or one of the other cell types in the organ. As is the data in Fig. 1e simply say that Svb is required for clonal survival.

We tested one additional RNAi line for Svb (**Figure Sup. 2b**), two for Ubr3 (**Figure Sup. 3d**), one for Yki (**Figure Sup. 5**), as well as an additional strong genomic allele for *svb* (**Figure Sup. 2f**). The results further confirm our initial data.

We now provide several pieces of evidence that only RNSCs have the capacity to proliferate and differentiate (**Figure Sup. 4c-c'**), as also supported by a recent study (Xu *et al.*, *Elife* 2018). We also add new analyses of MARCM *svb* clones' behavior, which strengthen our conclusions (**Figure Sup. 2g**).

4) There is variability in the length of RNAi knockdown in different experiments that raises some questions. For example, the Pri RNAi phenotype is quite a bit stronger than

the Svb RNAi phenotype: Pri kd eliminates RNSCs in 8 days but Svb RNAi takes 32 days to eliminate them. This could be a trivial consequence of kd efficiency, or it could argue that the Pri peptides have roles beyond Svb, which could profoundly affect how data is interpreted throughout the paper. Day8 Svb kd is used as the basis for modification/rescue by Yki. Does this look the same at 32 day kd when the Svb phenotype is stronger?

Although one cannot exclude additional roles of *pri* in adult RNSCs (as previously reported in embryonic tissues, see also response to Rev#1), we believe that results obtained for Ubr3 RNAi-mediated depletion (3 different lines) or genetic inactivation (null allele) are consistent with the interpretation. In addition, the effect of the constitutive repressor (OvoA) and now of the PPxY Svb mutant (**Figure 5e**) further support our conclusion.

Concerning genetic interactions, we have chosen 8 days of treatment precisely because it provides an intermediate phenotype well adapted to test for modifications (enhancement or suppression). Genetic assays using Gal4-driven manipulations have, however, obvious limitations. We now add compelling evidence of functional interactions between the Hippo pathway and Svb/Pri, with dose-sensitive genetic assays making use of genomic mutant alleles (**Figure Sup. 7 c,c',d**).

5) In a related concern, the rescue of *svb*-RNAi by UAS-p35 in Fig3c is important as it justifies the leap to apoptosis as the cause of RNSC loss. However, its done with 8day *svb* RNAi which is a much more subtle phenotype than the 32day RNAi shown in Fig 1. I would be much more convinced if p35 rescued the RNSC loss after 32 days of *svb* RNAi.

TUNEL assays now clearly show that the lack of *svb* in RNSCs induces apoptosis (**Figure 3c**). We also add data showing the rescue of *svb*-RNAi by DIAP1 at 32 days (**Figure 4a**).

6) The data implying a role for Svb in Yki phenotypes in eyes and discs is exciting, as it implies a potentially conserved relationship across tissues. However, the disc data is incomplete and inserted into supplemental data section. It should be strengthened and moved into the main body of the text: e.g. Figs S3D and E respectively show the effects of OvoA or Pri co-expression with Yki in the wing or eye.

The images in S3D nicely show OvoA shrinks Yki-driven wing growth; however, this would be much stronger if they could show this correlates with suppression of Yki driven expression of one of its targets. In the eye data in S3E, GMR-Pri has a hyperplastic phenotype on its own, but the combined GMR-yki/Pri eye seems to change shape more than size (flatter and more hammer-like). Organ shape is an overly crude way to measure effects on a putative shared Svb/Yki transcriptome. The authors should simply test whether coexpression of Pri or Svb enhances Yki-driven diap-lacZ (or ex-lacZ or bantam-lacZ) and reciprocally whether RNAi of Svb or Pri suppresses this effect.

We agree with the reviewer that testing Svb contribution to the expression of Yki target genes, in different contexts during development, is interesting. We actually performed a first set of experiments, which have yet been challenged by lethality issues of stocks combining the different mutations/transgenes. Exploring this question will thus require dedicated efforts (generation of recombined stocks etc...) that extend beyond the scope

of our current work.

Other comments:

1) Caspase-driven death is fairly rapid, but that latency to death following kd of Svb is on the order of days. Why is this? If Yki and Svb control Diap1 transcription, which interacts with and inhibits primed caspases, they should die fairly quickly. On a related note, overexpressed Rpr is not sufficient to kill ISCs (Jin et al, 2017, Stem Cell Reports; from the Edgar lab), so I'm a little concerned that the effect of Diap in RNSC survival may not be via the classic Rpr/Hid-Diap1-Caspase pathway.

We agree with the reviewer that adult intestine stem cells are quite resistant to cell death, and believe that our studies would help understanding this general feature of somatic stem cells. Our novel data (**Figure 3c**) show that the overexpression of Rpr strongly increases apoptosis, even though it requires several days of treatments, as observed for the knockdown of Svb and Ubr3 on the one hand, and of Yki or the overexpression of Hippo on the other.

2) In Figures 2, 3 and 4 the percent of surviving RNSCs is presented in a series of individual green bar graphs to the right of an optical slice. These quantitative data are critical and claim to show rescue and epistasis between *pri*, *hpo*, *yki* and *svb*, but they are hard to evaluate since they are not combined into one graph with stats & p-values. They should all be presented as in Fig. S3, panel F...why is this key data in supplement?

As requested, quantifications are now presented in a clearer way with n and p values in each figure and supplemental material.

7) The data in Fig. 3b on cell size data not meaningful vis a vis apoptosis or Hippo status. Apoptotic cells bleb and fragment. Are *svb* RNAi cells positive for tunel?

We agree and removed the incriminated data. Importantly, TUNEL assays no show that *svb*-mutant RNSCs undergo apoptosis (**Figure 3c**).

Reviewer #3

In this interesting manuscript, Bohere and colleagues present strong evidence that the transcription factor Shavenbaby (Svb) is expressed in the adult Renal/Nephritic Stem Cells (RNSCs) of *Drosophila melanogaster*, and it is critically required for their survival. More specifically, they relate Svb with the Hippo pathway, disclosing its physical interaction with Yorkie (Yki) which, together, positively regulate the expression of the apoptosis inhibitor DIAP1. In addition, they demonstrate that the processing of Svb by *pri* and *ubr3* is required for its correct function, in a similar way to other tissues already reported in the literature.

They show that loss of function of Svb (either with RNAi or the constitutive repressor OvoA) induces a serious loss in the population of RNSCs in the ureters and lower part of the Malpighian tubules. Furthermore, they determine the interaction of Svb and Yki, either by co-immuno-precipitation and epistatic experiments, demonstrating that their genetic and physical interaction is required for the correct expression of DIAP1, and therefore, inhibiting apoptosis in the tubules. The experimental data presented is elegantly convincing, and the conclusions fit in to the question initially asked.

Main points:

Are these really stem cells? Since Hou's paper (which may have issues), there has been a lack of evidence that these cells actually divide, and I notice that the authors had trouble finding evidence of mitosis in this MS. Alternatively, they have been described as myoendocrine by Garaoya and by Sozen, and as 'tiny cells' by the latter. I don't think this detracts from the publishability of this MS; but I think that, to prevent future embarrassment, the authors should at least put a sentence in the introduction acknowledging that their identity is not entirely settled. Better still, if they have evidence that these cells are mitotically competent, they should include it in this MS.

We thank the reviewer for her/his advice and modify the text accordingly (**P5 and references**). In addition to PH3 staining, we now provide strong evidence supporting the stemness of RNSCs, including G-TRACE lineage analysis (**Figure Sup. 4c,c'**) and a better description of MARCM clones in control and *svb*-depleted conditions (**Figure Sup. 1g**). Of note, a recent paper from an independent group (Xu *et al.*, Elife 2018) further corroborates this notion.

The authors identify the RNSCs by the GFP positive cells under the *esgGal4* driver, and they present a new protein expressed in these cells, Hindsight, as specific of these stem cells. Although it is interesting that this protein is specifically expressed in the RNSCs, it is not clear the relation that this protein has with them. In addition, the loss of GFP due to loss of RNSCs in some of the figures doesn't make clear the point that these cells are actually not there. Maybe it would be good to include a nuclear counterstaining (DAPI/To-Pro3) in, at least one figure, to show this (e.g. Fig1.c), as well as more information about Hindsight, or other staining that show the identity of RNSCs (maybe staining for Delta, or another stem cell marker, to give strength to the fact that these are RNSCs).

As suggested, we include additional data showing nuclear staining (**Figure 1c'**), a better description of Hindsight in RNSCs (co-localization with *esg*, *dome*-MESO, **Figure Sup.**

2c-e), as well as additional markers of stemness (Armadillo accumulation **Figure 1c'**, **Sup. 2a**) or Cut staining of MARCM clones, since RNSCs express low levels of Cut (Xu *et al.*, Elife 2018).

Minor points:

In the figure 4a, the authors show that OvoB is able to rescue the loss of RNSCs caused by the overexpression of Hpo (which, in theory, it will induce the degradation of more Yki, so it can be considered similar to *yki*-RNAi). However, this result doesn't seem to be in line with the phenotype shown in Suppl. Figure 3c, in which OvoB is not able to rescue *yki*-RNAi phenotype. It is true that the phenotype of the overexpression of Hpo (Figure 4a) is milder than the *yki*-RNAi (Suppl. Figure 3c), and the phosphorylation of Yki depends of other components of the Hippo pathway, but it is not clear the relation between these two apparently contradictory results.

If it is true that in mammals, LATS phosphorylation can prime YAP/TAZ for CK1d/e-mediated phosphorylation and b-TrCP-induced protein degradation [3,4]. In *Drosophila*, Wts-mediated phosphorylation mostly leads to Yki cytoplasmic retention by 14-3-3 proteins and subsequent downregulation of downstream targets. Therefore, elevated Hpo function does not necessarily mimic the genetic inactivation of Yki. In addition, we acknowledge that the corresponding sections of our ms was unclear. We therefore extensively rephrase the text (**P10-11**) to better present our results. We also add a whole set of new experiments to clarify the relationships between Svb and Yki. We show that Yki is able to bind both the activator and the repressor forms of Svb (**Figure 5c**). Mutations in the PPxY motifs of Svb show that they are required for the binding to Yki (**Figure 5c**). Importantly, the PPxY mutated form of Svb behaves *in vivo* as the repressor in RNSCs (**Figure 5d,e**), suggesting that the interaction with Yki may contribute to the activation of Svb in stem cells.

In the figure 4d and 4e, it is beautifully shown how DIAP1 is induced by Svb/Yki, and that the constitutive repressor OvoA is enough to block the expression of DIAP1 even when Yki is overexpressed. Have you tried to do the opposite experiment? Observe the expression of DIAP1 in a *yki*-RNAi/OvoB background?

As OvoB is not able to rescue *Yki*-RNAi, we guess that the expression of DIAP1 should be shut down as well.

In addition, it would be informative another experiment to show ongoing apoptosis of RNSCs (Caspase staining/western-blot, or TUNEL assay?), if possible.

Our previous attempts to document apoptosis in adult stem cells (staining for DCP1 or cleaved Caspase) have proven to be technically challenging. We agree that this point was important and TUNEL data (**Figure 3c**) now strongly reinforce our conclusion that RNSCs lacking *svb* undergo apoptosis.

Finally, in the overall manuscript it shows that the architecture of the tubule (at least the ureter) looks normal, even when the RNSC population is absolutely abolished (e.g. in Figure 2d). It looks surprising that the tubule – and the organism- survives without major defects even without these (stem?) cells. Have you observed any defects in later stages,

both morphologically or physiologically? (secretion rates, stress tolerance, or response to tissue damage?). In my opinion, this would strength the requirement for the RNSCs and their function in the adult tubule.

This an excellent point. Unfortunately, one experimental difficulty is the current lack of a genetic system allowing manipulation of stem cells specifically in renal tubules. We are working on that, but we aren't aware of an available system suitable to focus on RNSCs. As an alternative, Jerome has spent a two months stay in Julian DOW's lab and started functional analyses of secretion in dissected tissues. This is certainly a promising path, to be also complemented by additional approaches, and we believe that it extends beyond the scope of this paper.

REVIEWERS' COMMENTS:

Reviewer #1 (Remarks to the Author):

The authors have addressed all previous concerns and have nicely improved the manuscript. These are exciting results linking Shavenbaby and the Hippo-pathway to protecting adult stem cells from apoptosis.

The results are convincing, and the paper is well written.

Minor points:

- in Fig 2e and Fig 4b, the y-axis should be labelled with # of cells
- in Fig 3c, 'apoptosis index' should be corrected to 'apoptosis index', and '# of tunel positive cells per ROI' should be changed to '# of TUNEL-positive cells per ROI'

Reviewer #2 (Remarks to the Author):

I am satisfied that the cells in question display properties of stem cells and are eliminated by apoptosis in the background of Svb loss. The authors have also responded to review requests for enhanced mechanistic insight regarding the interaction between Svb and Yki. The genetic epistasis data are quite reasonable, although it remains somewhat unclear how Svb regulates Yki. Perhaps it enhances recruitment of cofactors on DNA? The addition of the PPxyY-to-A allele has also strengthened the argument that the Svb-Yki interaction has in vivo significance.

Reviewer #3 (Remarks to the Author):

I am now satisfied with the responses to my points.

RESPONSE TO REVIEWERS' COMMENTS

Reviewer #1:

The authors have addressed all previous concerns and have nicely improved the manuscript. These are exciting results linking Shavenbaby and the Hippo-pathway to protecting adult stem cells from apoptosis.

The results are convincing, and the paper is well written.

Minor points:

- in Fig 2e and Fig 4b, the y-axis should be labelled with # of cells
- in Fig 3c, 'apoptosis index' should be corrected to 'apoptosis index', and '# of tunel positive cells per ROI' should be changed to '# of TUNEL-positive cells per ROI'

We thank the reviewer for his scrutiny. All suggested modifications/corrections have been introduced in corresponding Figures.

Reviewer #2

I am satisfied that the cells in question display properties of stem cells and are eliminated by apoptosis in the background of Svb loss. The authors have also responded to review requests for enhanced mechanistic insight regarding the interaction between Svb and Yki. The genetic epistasis data are quite reasonable, although it remains somewhat unclear how Svb regulates Yki. Perhaps it enhances recruitment of cofactors on DNA? The addition of the PPxyY-to-A allele has also strengthened the argument that the Svb-Yki interaction has *in vivo* significance.

Reviewer #3

I am now satisfied with the responses to my points.